# Deterministic Differentiable Structured Pruning for Large Language Models

Weiyu Huang [* 1 2]  Pengle Zhang [* 1]  Xiaolu Zhang [2]  Jun Zhou [2]  Jun Zhu [1]  Jianfei Chen [1]

## Abstract

Structured pruning reduces LLM inference cost by removing low-importance architectural components. This can be viewed as learning a multiplicative gate for each component under an $\ell_0$ sparsity constraint. Due to the discreteness of the $\ell_0$ norm, prior work typically adopts stochastic hard-concrete relaxations to enable differentiable optimization; however, this stochasticity can introduce a train–test mismatch when sampled masks are discretized for deployment and restricts masks to a bounded, near-binary range. To address this, we propose Deterministic Differentiable Pruning (DDP), a mask-only optimization method that eliminates stochasticity by directly optimizing a deterministic soft surrogate of the discrete $\ell_0$ objective. Compared with prior approaches, DDP offers greater expressiveness, reduced train–test mismatch, and faster convergence. We apply our method to several dense and MoE models, including Qwen3-32B and Qwen3-30B-A3B, achieving a performance loss as small as 1% on downstream tasks while outperforming previous methods at 20% sparsity. We further demonstrate end-to-end inference speedups in realistic deployment settings with vLLM. Our code is publicly available.

## 1. Introduction

Large language models (LLMs) (Vaswani et al., 2017; Brown et al., 2020; Guo et al., 2025a) have demonstrated remarkable performance across a wide range of challenging tasks, including reasoning, code generation, and decision making. Despite these advances, deploying LLMs at scale remains resource-intensive, often requiring substantial compute, memory, and serving infrastructure. These costs pose a significant barrier to practical deployment, especially in budget-constrained settings.

Structured pruning (Ma et al., 2023; Zhang et al., 2024; Li et al., 2024) offers a promising way to reduce these costs by removing entire architectural components—e.g., attention heads, hidden dimensions, or MLP channels—to reduce model size and inference cost. Unlike unstructured sparsity (Frantar & Alistarh, 2023; Sun et al., 2023), which prunes individual weights and often requires specialized support for speedups, structured pruning is compatible with standard dense operators and existing hardware, enabling efficient execution without added system complexity.

Despite its appeal, most structured pruning methods for modern LLMs rely on efficient one-shot approaches (Li et al., 2024; Guo et al., 2025b) that select components using heuristic importance scores. While fast, these heuristics can be brittle and often incur substantial quality degradation under aggressive sparsity. In contrast, learning sparsity via end-to-end weight updates is typically prohibitive at LLM scale, leaving a gap between practicality and quality.

A key observation motivating our approach is that *mask-only* optimization provides a practical middle ground between heuristic one-shot pruning and full weight tuning. Here, a *mask* is a set of learnable gating variables that determine whether each architectural component (e.g., a head or channel) is kept or pruned. In our setting, *all pre-trained weights are frozen and only the masks are optimized*. This is feasible because the search space is much smaller—even smaller than LoRA modules: for DeepSeek-R1 (Guo et al., 2025a) with 685B parameters, the number of mask variables is only on the order of tens of millions, enabling scalable gradient-based optimization that converges within a small token budget (e.g., < 30M tokens). As a result, mask learning can discover higher-quality sparsity patterns than heuristic one-shot pruning without the cost of full-model training.

Prior approaches that learn sparsity masks (Guo et al., 2023; Xia et al., 2024; Liu et al., 2025) improve over heuristic pruning, but they do not fully meet our goal of lightweight, scalable mask-only optimization. First, they typically learn masks *together with* weight updates (full or LoRA fine-tuning), which increases training cost and often requires large-scale pretraining-style data for stable recovery. Second, they commonly rely on stochastic hard-concrete relaxations (Louizos et al., 2018); in a mask-only setting, this

*Equal contribution [1]Department of Computer Science and Technology, Tsinghua University, Beijing, China [2]Ant Group, Beijing, China. Correspondence to: Jianfei Chen <jianfeic@tsinghua.edu.cn>.

*Proceedings of the 43rd International Conference on Machine Learning*, Seoul, South Korea. PMLR 306, 2026. Copyright 2026 by the author(s).

constrains masks to a bounded, near-binary range and introduces sampling noise, leading to slower convergence and a potential train–test mismatch when deterministic masks are required for sparsity control and deployment.

To address these issues, we propose **D**eterministic **D**ifferentiable **P**runing (DDP), a lightweight mask-only structured pruning framework that learns structured sparsity patterns via gradient-based optimization. We formulate pruning as an $\ell_0$-regularized optimization problem and enforce the sparsity constraint with an augmented Lagrangian method (ALM). To handle the non-differentiability of the $\ell_0$ norm, DDP introduces a deterministic smooth surrogate that is annealed to a sharp $\ell_0$ objective during training, removing sampling noise and avoiding the train–test mismatch induced by stochastic masks. Furthermore, DDP decouples the mask values used in the forward pass from those used for regularization, expanding the effective mask value range beyond near-binary constraints and improving performance. In addition, DDP includes an explicit binarization loss that encourages mask polarization, leading to faster and more stable convergence. We further show that knowledge distillation integrates naturally into our framework with minimal additional overhead. We validate DDP across both dense and mixture-of-experts (MoE) models, scaling to flagship open-source models with tens of billions of parameters. Across multiple LLM families and standard benchmarks, DDP consistently delivers superior quality–efficiency trade-offs compared to prior state-of-the-art methods.

**Conflict of Interest**  The authors declare that they have no financial conflicts of interest related to this work.

## 2. Preliminary

This section provides a mathematical formulation of *mask optimization* for structured pruning. We first introduce a unified masking framework for pruning LLM submodules (Section 2.1), then cast targeted pruning as a constrained optimization problem and review how prior hard-concrete reparameterization methods address it (Section 2.2). We further analyze the objective mismatch of these approaches in the context of structured LLM pruning (Section 2.3).

### 2.1. Unified Masking Formulation

Transformer-based LLMs are composed of repeated submodules such as multi-head attention and MLPs, whose outputs can be written as a sum of $K$ independent components (e.g., attention heads or intermediate channels). We model structured pruning by attaching a multiplicative mask to each component; given input $\mathbf{X}$, the module output is

$$\mathbf{y} \;=\; \sum_{k=1}^{K} m_k \, \mathbf{f}_k(\mathbf{X}), \qquad m_k \in \mathbb{R}, \qquad (1)$$

where $\mathbf{f}_k(\cdot)$ is the $k$-th component and $m_k$ is the corresponding gate ($m_k = 0$ prunes; $m_k \neq 0$ keeps). For dense LLMs, we apply this to both attention and MLP blocks.

**Multi-head attention.**  For a multi-head attention layer with $H$ heads, we take $K = H$ and define the contribution of the $h$-th head as

$$\mathbf{f}_h^{\text{attn}}(\mathbf{X}) = \text{Attn}\!\left(\mathbf{X}\mathbf{W}_Q^{(h)}, \; \mathbf{X}\mathbf{W}_K^{(h)}, \; \mathbf{X}\mathbf{W}_V^{(h)}\right) \mathbf{W}_O^{(h)}, \; (2)$$

where $\text{Attn}(\cdot)$ is the scaled dot-product attention.

**MLP channels.**  For a (gated) MLP with intermediate width $C$, we set $K = C$ and write

$$\mathbf{f}_j^{\text{mlp}}(\mathbf{X}) = \left(\varphi(\mathbf{X}\mathbf{u}_j) \odot (\mathbf{X}\mathbf{g}_j)\right) \mathbf{v}_j, \qquad (3)$$

where $\mathbf{u}_j, \mathbf{g}_j$ are the $j$-th up/gate columns, $\mathbf{v}_j$ is the corresponding down-projection row, and $\varphi(\cdot)$ denotes the element-wise nonlinearity (e.g., GELU).

Given an $L$-layer model, the mask variables are summarized in Table 1.

*Table 1.* Mask variables for attention and MLP pruning.

| Submodule | Attention | MLP |
|---|---|---|
| Masks | $\boldsymbol{m}^{\text{attn}} \in \mathbb{R}^H \; (\times L)$ | $\boldsymbol{m}^{\text{mlp}} \in \mathbb{R}^C \; (\times L)$ |

### 2.2. Mask Optimization by $\ell_0$ Regularization

Our goal is to optimize the collection of *structured* masks $M = \{\boldsymbol{m}^{\text{attn}}, \boldsymbol{m}^{\text{mlp}}\}$ that minimize the language modeling objective while meeting a prescribed sparsity budget. For notational convenience, we drop the explicit set notation and use $\boldsymbol{m}$ to denote a generic structured mask in $M$. A natural starting point is $\ell_0$-regularized training,

$$\min_{\boldsymbol{m}} \; \mathcal{L}_{\text{ce}}(\theta, \boldsymbol{m}) \;+\; \lambda \|\boldsymbol{m}\|_0, \qquad (4)$$

where $\mathcal{L}_{\text{ce}}$ denotes the cross-entropy loss and $\|\boldsymbol{m}\|_0$ is the $\ell_0$-norm, which counts the number of active components.

In practice, deployment often requires an *explicit* sparsity level rather than tuning a regularization strength $\lambda$. Therefore, Equation (4) can be further cast into a constrained optimization problem,

$$\min_{\boldsymbol{m} \in \mathcal{S}} \; \mathcal{L}_{\text{ce}}(\theta, \boldsymbol{m}), \qquad (5)$$

where $\mathcal{S}$ denotes the target constraint set. In particular, we enforce a targeted *keep ratio* $\rho$ for $\boldsymbol{m}$ (equivalently, a pruning/sparsity ratio of $\alpha \triangleq 1 - \rho$)

$$\bar{m} \triangleq \frac{1}{K}\|\boldsymbol{m}\|_0 \;=\; \frac{1}{K}\sum_{k=1}^{K}\|m_k\|_0 \;=\; \rho, \qquad (6)$$

where $\bar{m}$ denotes the average keep ratio of $\|\boldsymbol{m}\|_0$.

Directly solving Equation (5) is difficult in the large-scale LLM setting because current optimization methods are based on gradient descent and are tailored to unconstrained, smooth objectives. A common remedy is to convert the constraint into an unconstrained objective via an augmented Lagrangian method (ALM). Specifically, we add a sparsity penalty for each $\boldsymbol{m}$ that takes the form:

$$\mathcal{L}_{\text{sparsity}}(\|\boldsymbol{m}\|_0) \triangleq \lambda_1(\bar{m} - \rho) + \lambda_2(\bar{m} - \rho)^2, \quad (7)$$

where $\lambda_1$ and $\lambda_2$ act as the Lagrange multiplier and quadratic penalty, respectively.

**Hard-Concrete Relaxation.** While the augmented Lagrangian enforces an explicit sparsity constraint, the fundamental difficulty remains: the $\ell_0$ regularizer is discrete and non-differentiable. To enable gradient-based optimization, prior work (Fang et al., 2024; Wang et al., 2020; Xia et al., 2024) adopts the hard-concrete relaxation (Louizos et al., 2018), redefining $\boldsymbol{m}$ as a $[0, 1]$-valued *random* mask variable to model $\ell_0$ norm. In the forward pass, masks $\boldsymbol{m}$ are generated using the hard-concrete mapping, with randomness injected via auxiliary variable $\boldsymbol{u}$, we define the projection $\boldsymbol{m} = \Phi(\boldsymbol{z}, \boldsymbol{u})$ as:

$$\begin{aligned} \boldsymbol{u} \sim U(0, 1), \quad \boldsymbol{v} &= \sigma\big(\log \boldsymbol{u} - \log(1 - \boldsymbol{u}) + \boldsymbol{z}\big), \\ \bar{\boldsymbol{v}} = \boldsymbol{v}\,(r - l) + l, \quad \boldsymbol{m} &= \text{Clamp}(\bar{\boldsymbol{v}}, 0, 1). \end{aligned} \quad (8)$$

where $\sigma(\cdot)$ denotes the sigmoid function, $l < 0 < 1 < r$ are stretching parameters that concentrate probability mass at $0, 1$, and $\boldsymbol{z}$ denotes the corresponding latent parameter to be optimized. This relaxation admits a closed-form surrogate for the expected $\ell_0$ norm:

$$\mathbb{E}\big[\|\boldsymbol{m}\|_0\big] = \sum_{k=1}^{K} \mathbb{P}(m_k > 0) = \sum_{k=1}^{K} \sigma(z_k - \log(-l/r)). \quad (9)$$

Since the mask $\boldsymbol{m}$ is now a random variable, the expected keep ratio $\mathbb{E}\big[\|\boldsymbol{m}\|_0\big]$ can replace $\|\boldsymbol{m}\|_0$ in the augmented Lagrangian penalty in Equation (7) to enforce the target sparsity level.

### 2.3. Drawbacks of Hard-Concrete Relaxation

The resulting objective used by hard-concrete methods is

$$\min_{\boldsymbol{z}} \; \mathbb{E}_{\boldsymbol{u} \sim U(0,1)}\Big[\mathcal{L}_{\text{ce}}\big(\theta, \boldsymbol{m}\big) + \mathcal{L}_{\text{sparsity}}\big(\|\boldsymbol{m}\|_0\big)\Big], \quad (10)$$

where $\boldsymbol{m} = \Phi(\boldsymbol{z}, \boldsymbol{u})$ is the hard-concrete mapping from noise $\boldsymbol{u}$ and logits $\boldsymbol{z}$ to a bounded mask.

We discover two main drawbacks in this formulation. **(1) Train–Test Mismatch.** While masks $\boldsymbol{m}$ are sampled during training, evaluation and deployment require deterministic masks. Converting these random variables into discrete values at inference creates a train–test mismatch, which can lead to unstable sparsity enforcement and degraded performance. Moreover, by introducing randomness into training, the optimization objective becomes an expectation over random variable $\boldsymbol{u}$. This introduces noise into both the forward and backward passes, causing slow convergence. **(2) Limited mask expressivity.** In the unified masked formulation (Equation (4)), the masking variables $\boldsymbol{m}$ are naturally real-valued when calculating language modeling loss $\mathcal{L}_{\text{ce}}$. However, the hard-concrete mapping constrains $\boldsymbol{m}$ to a bounded, near-binary range during training, shrinking the effective search space and potentially hindering the discovery of high-quality sparsity patterns.

## 3. Method

To address these issues, we introduce Deterministic Differentiable Pruning (DDP), a fully deterministic mask-only optimization framework that operates over a continuous mask space. We present the method in Section 3.1, followed by a convergence analysis in Section 3.2 and several practical extensions in Section 3.3. An overview of the method is shown in Figure 1.

### 3.1. Deterministic Differentiable Pruning

First, given the latent real-valued parameter $\boldsymbol{z}$, we replace the stochastic hard-concrete sampling in Equation (8) with a deterministic ReLU gate in the forward pass:

$$\boldsymbol{m} = \text{ReLU}(\boldsymbol{z}). \quad (11)$$

This expands the forward mask space from near binary values to $m_k \in [0, \infty)$, enabling continuous scaling of component contributions while avoiding negative mask values that could induce sign flips and undesired cancellation.

Furthermore, to address the non-differentiability of the $\ell_0$ norm *without* introducing stochastic sampling, we construct an annealed soft surrogate that is used when computing the augmented Lagrangian sparsity term in Equation (7). Rather than regularizing the forward mask $\boldsymbol{m}$ directly, we apply a deterministic mapping that projects logits $\boldsymbol{z}$ to *retention surrogate scores* $\boldsymbol{s} \in [0, 1]$, where each $s_k$ measures how strongly the $k$-th component is retained. At iteration $t$, the mapping $\boldsymbol{s} = \phi(\boldsymbol{z}; \mu_t)$ is defined as

$$\begin{aligned} \boldsymbol{v} = \sigma\Big((\boldsymbol{z} - \mu_t)\tfrac{C_0}{\mu_t}\Big), \quad \bar{\boldsymbol{v}} &= \boldsymbol{v}\,(r - l) + l, \\ \boldsymbol{s} &= \text{Clamp}(\bar{\boldsymbol{v}}, 0, 1). \end{aligned} \quad (12)$$

where $\mu_t$ is an annealed sharpness parameter controlling how quickly the mapping transitions from 0 to 1. We fix $l = -0.1$, $r = 1.1$, and $C_0 \approx 2.4$ such that $s(0) = 0$ and

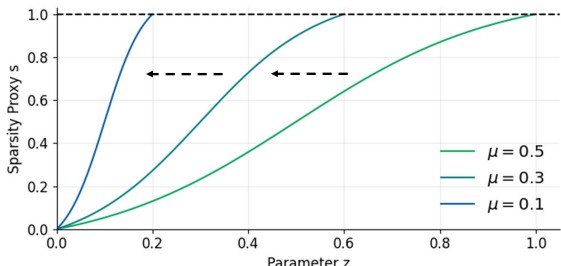

Figure 1. **Deterministic Differentiable Pruning overview. Left:** Masked formulation for dense and MoE models. For dense models, we prune attention heads and MLP channels; for MoE models, we prune expert channels only. **Right:** Mask-only optimization with decoupled forward masks and retention scores for regularization, enabling deterministic training and an expanded mask range.

Figure 2. Deterministic surrogate mapping in DDP. Annealing $\mu$ sharpens the soft sigmoid projection, progressively approximating $\ell_0$ regularization.

$s(2\mu_t) = 1$. In particular, for any $z_k \geq 2\mu_t$, the score saturates to $s_k = 1$, and the corresponding component is treated as fully retained in regularization.

The resulting $s$ are then used in the Lagrangian regularization term to enforce the target keep ratio $\rho$. Denote $\bar{s} = \frac{1}{K}\sum_k s_k$ as their mean. The penalty is given as:

$$\mathcal{L}_{\text{sparsity}}(\boldsymbol{s}) = \lambda_1(\bar{s} - \rho) + \lambda_2(\bar{s} - \rho)^2. \qquad (13)$$

Let $T$ denote the total number of training steps. During training, we anneal $\mu_t$ according to

$$\mu_t = \mu_0 - (\mu_0 - \mu_T)\sqrt{\frac{t}{T}}. \qquad (14)$$

We set $\mu_0 = 0.5$ and initialize $z_k = 1$ for all $k$, corresponding to a dense forward pass at the start of training, and then anneal $\mu_t$ close to 0. As illustrated in Figure 2, the mapping becomes increasingly sharp over time, so the surrogate mapping falls back to exact $\ell_0$ behavior when $\mu_t \to 0$. We implement clamping and ReLU with a straight-through estimator (STE) to preserve gradients.

Finally, while Equation (13) enforces the *average* keep ratio, it does not by itself encourage individual retention scores to

become decisively binary when $\mu_t$ is still large. To accelerate convergence, we introduce an additional binarization regularizer on $\{s_k\}_{k=1}^K$:

$$\mathcal{L}_{\text{bin}}(\boldsymbol{s}) = \lambda_3 \frac{1}{K}\sum_{k=1}^K s_k(1 - s_k), \qquad (15)$$

which mostly penalizes intermediate values and is minimized at endpoints, thereby pushing each $s_k$ toward $\{0, 1\}$. Combined with the deterministic surrogate mapping, this term stabilizes optimization by encouraging ambiguous components to commit early. In practice, we update the $\lambda$ coefficients via gradient ascent alongside the mask parameters so as to enforce the target sparsity constraint during training. Finally, after training we prune components whose mask values are zero and fold the remaining nonzero masks into the model parameters for deployment. We summarize our workflow in Section A.

### 3.2. Analysis

Our final training objective is

$$\min_{\boldsymbol{z}} \mathcal{L}_{\text{ce}}(\theta, \boldsymbol{m}) + \mathcal{L}_{\text{sparsity}}(\boldsymbol{s}) + \mathcal{L}_{\text{bin}}(\boldsymbol{s}), \qquad (16)$$

where we optimize mask logits $\boldsymbol{z}$ with a deterministic forward gate $\boldsymbol{m} = \text{ReLU}(\boldsymbol{z})$ and annealed retention scores $\boldsymbol{s} = \phi(\boldsymbol{z}; \mu)$ enforcing the keep budget. In the appendix, we show the following guarantee. For any fixed annealing level $\mu$, the updates used in our implementation define a fully deterministic constrained optimization procedure for the relaxed constraint $\sum_k s_k = P$ (with $P = \lceil \rho K \rceil$): under mild conditions, the resulting iterates admit accumulation points that are feasible and first-order stationary. Moreover, when we run the procedure in stages with $\mu \downarrow 0$ and the binarization term drives $\boldsymbol{s}$ toward $\{0, 1\}^K$, the relaxed budget $\sum_k s_k = P$ becomes equivalent to the discrete hard budget induced by the forward mask, $\sum_k \mathbb{I}[z_k > 0] = P$.

Consequently, for sufficiently large stages $r$, DDP outputs masks that satisfy the target $\ell_0$ keep budget *exactly*, while each stage solution is a Karush–Kuhn–Tucker (KKT) point of the corresponding relaxed constrained surrogate.

**Theorem 3.1** (Exact budget recovery (informal)). *Consider the staged optimization of Equation* (16) *with annealing* $\mu \downarrow 0$. *Under certain assumptions, for each stage* $r = 1, 2, 3..$ *(with fixed* $\mu_r$*), the deterministic augmented-Lagrangian updates generate iterates whose accumulation points are (i) feasible for the relaxed equality constraint* $\sum_k s_k = P$, *and (ii) first-order stationary (KKT) for the corresponding relaxed constrained surrogate under the same STE gradients used in practice. If in addition the binarization loss drives* $\boldsymbol{s}$ *near* $\{0, 1\}^K$ *and the logits become separated from the ReLU threshold for large* $r$, *then for sufficiently large stages the KKT points produce masks that satisfy the* exact *hard budget*

$$\sum_{k=1}^{K} \mathbb{I}[z_k > 0] = P.$$

Proofs and detailed analysis are in Section B.

### 3.3. Extensions

**Distillation.** Since we keep the pre-trained weights fixed and optimize only the masks, the dense model serves as a parameter-free teacher. This incurs no extra optimizer memory and requires only two forward passes. We define the KL divergence between the teacher and student as:

$$\mathcal{L}_{\mathrm{kl}}(\boldsymbol{m}, \theta) = \sum_i D_{\mathrm{kl}}(P_t(\mathbf{X}, i) \,\|\, P_s(\mathbf{X}, i)), \qquad (17)$$

where $P_t(\mathbf{X}, i)$ and $P_s(\mathbf{X}, i)$ are the teacher/student next-token distributions over the vocabulary at position $i$, respectively. We add the distillation loss to $\mathcal{L}_{\mathrm{ce}}$ during training.

**MoE.** For Mixture-of-Experts (MoE) models, since most parameters reside in the expert MLPs, we apply pruning only to the experts and keep the attention blocks unchanged. We use the same channel-wise decomposition within each expert and weight expert outputs by the router scores $\pi_e(\mathbf{X})$:

$$\mathrm{MoE}(\mathbf{X}) = \sum_{e=1}^{E} \pi_e(\mathbf{X}) \sum_{j=1}^{C} m_{e,j}\, \mathbf{f}_{e,j}^{\mathrm{mlp}}(\mathbf{X}), \qquad (18)$$

where $E$ is the number of experts and $C$ is the intermediate channel dimension per expert. Accordingly, the mask parameters for a single MoE layer are $\boldsymbol{m}^{\mathrm{moe}} \in \mathbb{R}^{E \times C}$.

**Fine-grained Sparsity Control.** Deterministic Differentiable Pruning supports different sparsity granularities by applying the sparsity regularizer in Equation (13) at the group level. We can partition mask entries into groups $\mathcal{G}$ (e.g., per layer or per expert) and enforce a per-group keep-ratio budget:

$$\mathcal{L}_{\mathrm{sparsity}}(\boldsymbol{s}) = \frac{1}{|\mathcal{G}|} \sum_{g \in \mathcal{G}} \left[ \lambda_1 \big(\bar{s}_g - \rho\big) + \lambda_2 \big(\bar{s}_g - \rho\big)^2 \right], \ (19)$$

where $\bar{s}_g = \frac{1}{|g|} \sum_{k \in g} s_k$ is the average retention score in group $g$. This enables uniform sparsity across layers or within each MoE expert, and often improves speedups by yielding more regular, hardware-friendly patterns.

## 4. Related Work

Based on how pruning decisions are obtained and whether model weights are updated, existing methods can be grouped into (i) one-shot pruning, (ii) sparsity-aware training, and (iii) mask-only optimization.

**One-shot Pruning** One-shot pruning methods remove structures using heuristic importance criteria. For dense LLMs, LLM-Pruner (Ma et al., 2023) scores weights or modules via gradient-based sensitivity, while Lo-RAPrune (Zhang et al., 2024) estimates importance through lightweight LoRA (Hu et al., 2022) updates. LoRAP (Li et al., 2024) further combines a gradient-free channel criterion with low-rank attention compression, and Slim-LLM (Guo et al., 2025b) prunes attention heads and MLP channels using similarity-based scores. For MoE architectures, NAEE (Lu et al., 2024) prunes experts via (approximate) search over expert combinations, while $D^2$-MoE (Gu et al., 2025) constructs shared experts via weighted merging. Camera (Xu et al., 2025) and HEAPr (Li et al., 2025b) treat expert channels as prunable units and rely on Hessian- or activation-based importance to identify redundancy. Though fast, these methods can incur larger performance drops because they rely on handcrafted metrics.

**Sparsity-aware Training** Another line of work learns sparsity patterns while updating model weights through loss-based training, often using hard-concrete relaxations (Louizos et al., 2018) to enable gradient-based mask learning. For example, ShearedLLaMA (Xia et al., 2024) learns pruning masks during continued training, while Compresso (Guo et al., 2023) and PAT (Liu et al., 2025) combine mask learning with LoRA-based fine-tuning. Compared with mask-only optimization, these approaches typically require orders of magnitude more training compute due to more trainable parameters and larger token budgets. For semi-structured setting, AST (Huang et al., 2025a) and CAST (Huang et al., 2025b) achieve lossless 2:4 models on LLaMA2 and LLaMA3 models.

**Mask-only Optimization** Relatively few works focus on mask-only optimization. Recently, MaskLLM (Fang et al.,

*Table 2.* Performance of pruned dense LLMs under different methods. All baseline methods are evaluated after fine-tuning. **Bold** indicates the best performance at the same pruning ratio.

| Model | LLaMA-7B | | | | LLaMA-2-7B | | | | LLaMA-13B | | | |
|---|---|---|---|---|---|---|---|---|---|---|---|---|
| Sparsity Ratio | 20% | | 50% | | 20% | | 50% | | 20% | | 50% | |
| Method | Wiki2↓ | Mean Acc↑ | Wiki2↓ | Mean Acc↑ | Wiki2↓ | Mean Acc↑ | Wiki2↓ | Mean Acc↑ | Wiki2↓ | Mean Acc↑ | Wiki2↓ | Mean Acc↑ |
| Original (Dense) | 12.62 | 65.96 | 12.62 | 65.96 | 12.18 | 66.63 | 12.18 | 66.63 | 11.58 | 68.79 | 11.58 | 68.79 |
| LoRAPrune | 16.80 | 60.05 | 30.12 | 49.71 | – | – | – | – | – | – | – | – |
| LoRAP | 16.35 | 61.70 | 30.90 | 52.17 | 14.67 | 61.20 | **26.26** | 52.31 | 13.58 | 64.71 | 22.66 | 58.20 |
| SlimLLM | 15.55 | 62.41 | 26.71 | 53.16 | 15.28 | 61.70 | 27.29 | 52.02 | 13.35 | 64.21 | 25.64 | 54.75 |
| Ours | **15.20** | **64.13** | **26.70** | **56.07** | **14.39** | **64.82** | 26.34 | **56.70** | **12.71** | **66.89** | **20.32** | **62.14** |

*Table 3.* Performance of pruned Mixture-of-Experts LLMs under different methods. **Bold** indicates the best performance at the same pruning ratio.

| Model | DeepSeekMoE-16B-Base | | | | | | Qwen3-30B-A3B | | | | | |
|---|---|---|---|---|---|---|---|---|---|---|---|---|
| Sparsity Ratio | 20% | | 40% | | 60% | | 20% | | 40% | | 60% | |
| Method | C4↓ | Mean Acc↑ | C4↓ | Mean Acc↑ | C4↓ | Mean Acc↑ | C4↓ | Mean Acc↑ | C4↓ | Mean Acc↑ | C4↓ | Mean Acc↑ |
| Original (Dense) | 9.05 | 62.28 | 9.05 | 62.28 | 9.05 | 62.28 | 12.13 | 71.08 | 12.13 | 71.08 | 12.13 | 71.08 |
| NAEE | 10.07 | 60.51 | 12.80 | 54.94 | 29.44 | 45.28 | 12.44 | 69.94 | 13.87 | 65.77 | 19.37 | 57.13 |
| $D^2$-MoE | 12.62 | 58.97 | 17.22 | 54.32 | 34.54 | 46.72 | 18.52 | 66.35 | 35.48 | 62.77 | 68.36 | 52.02 |
| Camera-P | 9.84 | 61.03 | 11.68 | 58.58 | 18.10 | 51.62 | 12.25 | 69.94 | 15.58 | 67.46 | 24.48 | 59.03 |
| HEAPr | 9.85 | 60.62 | 12.34 | 57.70 | 21.25 | 51.38 | - | - | - | - | - | - |
| Ours | **9.38** | **61.84** | **10.75** | **60.38** | **12.65** | **58.18** | **11.58** | **70.04** | **12.10** | **67.65** | **13.54** | **63.35** |

2024) explores this setting for $N : M$ sparsity using modified hard-concrete relaxations (Louizos et al., 2018).

## 5. Experiments

### 5.1. Experimental Settings

**Models and Setup**   We evaluate our pruning method on a diverse set of model families. We report results on LLaMA (Touvron et al., 2023) and Qwen3 (Yang et al., 2025) model families as dense baselines. For MoE baselines, we include DeepSeekMoE-16B (Dai et al., 2024) and Qwen3-30B-A3B (Yang et al., 2025). We define sparsity ratio $\alpha = 1 - \rho$ as the fraction of pruned *units* (attention heads and MLP channels for dense models; expert channels for MoE models).

**Evaluation**   We evaluate on a broad suite of downstream benchmarks. For dense models, we report WikiText-2 (Merity et al., 2017) perplexity. For MoE models, since some baselines are calibrated on WikiText, we instead evaluate perplexity on C4 (Raffel et al., 2020) to avoid potential train–test overlap. For zero-shot evaluation, we use EleutherAI's LM Evaluation Harness (Gao et al., 2021) on ARC-Easy and ARC-Challenge (Clark et al., 2018), OpenBookQA (Mihaylov et al., 2018), WinoGrande (Sakaguchi et al., 2021), PIQA (Bisk et al., 2020), HellaSwag (Zellers et al., 2019), MathQA (Amini et al., 2019), RTE (Wang et al., 2018), and BoolQ (Clark et al., 2019).

**Implementation Details**   Unless otherwise specified, we train masks on a 30M-token subset of FineWeb-Edu (Penedo et al., 2024). Experiments are run on 4 NVIDIA H20 GPUs. Hyperparameters are robust across different model families and sparsity ratios, so we use the same configuration for

all runs. Since sparsity is enforced through a training objective, the achieved sparsity can deviate slightly from the target; in all experiments, we keep the difference within 1%. Additional details are provided in Section C.

**Baseline Selection**   We focus on strong one-shot structured pruning baselines for fair and practical comparison. For dense LLMs, most one-shot methods rely on post-pruning fine-tuning with LoRA to recover accuracy; we therefore include LoRAP (Li et al., 2024), LoRAPrune (Zhang et al., 2024), and SlimLLM (Guo et al., 2025b), and report their results under a matched tuning token budget comparable to our mask-learning cost (30M tokens). For MoE models, prior structured pruning baselines are typically training-free and do not apply fine-tuning; accordingly, we compare against the strongest reported training-free methods, including NAEE (Lu et al., 2024), $D^2$-MoE (Gu et al., 2025), Camera-P (Xu et al., 2025), and HEAPr (Li et al., 2025b). We further include comparison with hard-concrete relaxation in the ablation section. We do not compare against weight updating training methods, as they typically use substantially more compute (e.g., ~50B tokens for ShearedLLaMA (Xia et al., 2024)), making them less comparable to our lightweight setting. In practice, our method requires only ~40 minutes per run for LLaMA-7B and ~60 minutes for DeepSeekMoE-16B.

### 5.2. Main Results

#### 5.2.1. DENSE LLMS

Table 2 reports pruning results on dense LLaMA models using WikiText-2 perplexity and mean zero-shot accuracy. Across model sizes and sparsity levels, DDP achieves the

*Table 4.* **Ablation on mask-learning components at 20% sparsity.** We report perplexity and zero-shot mean accuracy on LLaMA-7B and DeepSeekMoE-16B. HC: stochastic hard-concrete; Det. HC: deterministic hard-concrete; +EM: expanded-mask parameterization. **Bold** indicates the best performance among pruned variants at the same sparsity.

| | LLaMA-7B | | DeepSeekMoE-16B | |
|---|---|---|---|---|
| Method | PPL↓ | Mean Acc↑ | PPL↓ | Mean Acc↑ |
| Dense (0%) | 12.62 | 65.96 | 9.05 | 62.28 |
| HC | 16.52 | 59.95 | 9.55 | 61.58 |
| Det. HC | 16.30 | 61.74 | 9.52 | 61.65 |
| Det. HC + EM | 15.36 | 63.92 | 9.42 | 61.80 |
| **Ours** | **15.20** | **64.13** | **9.38** | **61.84** |

best or near-best performance. On LLaMA-7B, it improves mean accuracy from 62.41 to 64.13 at 20% sparsity and from 53.16 to 56.07 at 50%, while also reducing perplexity. On LLaMA-13B, the gains are larger under aggressive pruning, improving mean accuracy by $> 7$ points at 50% sparsity over baselines. Overall, these results highlight the benefit of directly optimizing masks, compared to one-shot pruning approaches followed by fine-tuning.

### 5.2.2. MIXTURE-OF-EXPERTS LLMs

Table 3 summarizes MoE pruning results using C4 perplexity and mean zero-shot accuracy. Across both DeepSeekMoE-16B and Qwen3-30B-A3B, Deterministic Differentiable Pruning (DDP) achieves the best accuracy and lowest perplexity at every sparsity level. The advantage grows with sparsity: on DeepSeekMoE-16B at 60% sparsity, we outperform the strongest baseline by +6.6 mean-accuracy points (58.18 vs. 51.62) while substantially reducing perplexity (12.65 vs. 18.10). Qwen3-30B-A3B exhibits the same trend, with DDP remaining notably more stable under aggressive pruning. Full results as well as performance on additional models are provided in Section D.

### 5.3. Ablation Study

#### 5.3.1. ABLATION ON INDIVIDUAL COMPONENTS

We conduct an ablation study to quantify the contribution of each component in our method. Relative to the standard hard-concrete relaxation, our approach introduces three key changes: (i) removing the stochastic term and enabling deterministic mask optimization with binarization loss; (ii) using ReLU gating in the forward pass for an expanded, more expressive mask parameterization; and (iii) the use of knowledge distillation.

Table 4 quantifies the contribution of each design choice at 20% sparsity on LLaMA-7B and DeepSeekMoE-16B. Starting from hard-concrete (HC), we obtain a deterministic variant by removing the sampling noise $u$ in Equation (8) during the forward pass and using the deterministic mask

*Table 5.* **Sparsity granularity ablation at 20% sparsity.** We report the change in perplexity and zero-shot mean accuracy relative to global sparsity for LLaMA-7B and DeepSeekMoE-16B.

| | LLaMA-7B | DeepSeekMoE-16B | |
|---|---|---|---|
| Metric | Layer-wise | Layer-wise | Expert-wise |
| Δ Perplexity ↓ | +0.04 | +0.47 | +1.17 |
| Δ Mean Acc. (pp) ↑ | -0.48 | -1.09 | -1.97 |

value in the regularizer, together with a binary regularizer to encourage discrete masks. This change yields consistent gains on both dense and MoE models, suggesting that sampling noise can hinder stable mask optimization. Introducing the expanded mask parameterization (+EM) further improves perplexity and accuracy, highlighting the benefit of decoupling forward gating from sparsity control. Finally, combining deterministic optimization, expanded masks, and distillation achieves the best performance across both models, substantially narrowing the gap to dense baselines and indicating that these components contribute additively.

#### 5.3.2. SPARSITY GRANULARITIES

We primarily report results under global sparsity for comparability with prior work; here we summarize the impact of using alternative granularities in Table 5. For dense models, we evaluate layer-wise uniform sparsity, while for MoE models we consider both layer-wise and expert-wise sparsity. Layer-wise sparsity causes only a small degradation on LLaMA-7B, but both alternatives lead to larger drops on DeepSeekMoE-16B, with expert-wise sparsity being the most restrictive, suggesting that global sparsity is particularly well-suited for MoE models because it can adapt to highly skewed expert activation frequencies, preserving frequently used experts while pruning rarely routed ones more aggressively. Overall, global sparsity yields the best accuracy, while finer-grained constraints provide a practical trade-off when hardware-friendly or deployment-specific sparsity patterns are required.

#### 5.3.3. TOKENS

We study how the token budget for mask learning affects performance. We run our method on LLaMA-7B and DeepSeekMoE-16B at 20% target sparsity and track both perplexity and zero-shot accuracy as we increase the number of training tokens. As shown in Figure 3, both metrics converge within a 60M-token budget, which we attribute to the low-dimensional nature of structured sparsity decisions. Since mask learning saturates quickly with minimal token budget, additional gains can be obtained by seamlessly transitioning to continued training on the pruned model. Notably, zero-shot performance recovers rapidly and saturates within ∼10M tokens for both models, whereas perplexity improves more gradually and continues to decrease up to

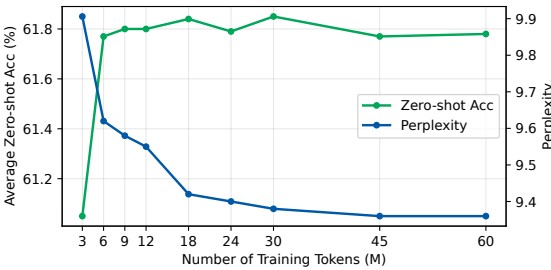

*(a)* DeepSeekMoE-16B at 20% sparsity.

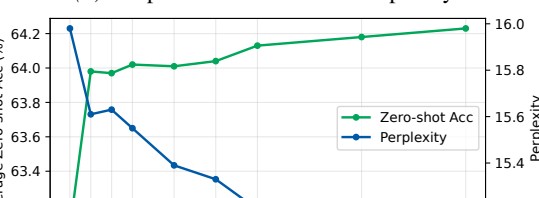

*(b)* LLaMA-7B at 20% sparsity.

*Figure 3.* Effect of different training tokens on perplexity and zero-shot mean accuracy on different models.

*Table 6.* **Dataset ablation at 20% sparsity.** We report the change in WikiText-2 perplexity and zero-shot mean accuracy relative to training masks on our default dataset (FineWeb-Edu) for LLaMA-7B and DeepSeekMoE-16B. Negative ∆PPL and positive ∆Acc indicate improvement.

| Metric | LLaMA-7B | | DeepSeekMoE-16B | |
|---|---|---|---|---|
| | **C4** | **LaMini** | **C4** | **LaMini** |
| ∆ WikiText-2 PPL ↓ | +0.35 | -0.10 | +0.21 | +0.11 |
| ∆ Mean Acc. (pp) ↑ | -1.14 | -0.53 | -0.68 | -0.87 |

60M tokens. We attribute this gap to knowledge distillation: it provides task-agnostic guidance that quickly preserves high-level capabilities and downstream generalization, while further perplexity reductions benefit from longer optimization to better match token-level distributions.

### 5.3.4. DATASETS

We ablate the dataset used for mask learning by training on either a weaker general pretraining corpus (C4) or an instruction-tuned corpus (LaMini). As shown in Table 6, both alternatives generally degrade performance, suggesting that dataset mismatch can harm the learned sparsity pattern. In particular, instruction-style data may bias masks toward behaviors that do not transfer to pretraining-style evaluation. Overall, these results indicate that high-quality pretraining-like data is preferred for mask learning.

### 5.4. Mask-Only Optimization vs. LoRA Recovery

We further compare DDP with a common two-stage alternative: training-free pruning followed by LoRA fine-tuning.

*Table 7.* Comparison between mask-only optimization and training-free pruning followed by LoRA recovery. PPL is evaluated on WikiText-2 for LLaMA-2-7B and on C4 for DeepSeekMoE-16B. DDP uses 30M mask-optimization tokens and keeps pretrained weights frozen.

| Model | Method | Tokens | PPL ↓ | Mean Acc. ↑ |
|---|---|---|---|---|
| | LoRAP | 0M | 15.02 | 59.44 |
| | LoRAP | 30M | 14.69 | 61.24 |
| LLaMA-2-7B | LoRAP | 60M | 14.48 | 62.77 |
| | LoRAP | 120M | 14.46 | 62.98 |
| | DDP (Ours) | 30M | **14.39** | **64.82** |
| | HEAPr | 0M | 9.85 | 60.62 |
| | HEAPr | 30M | 9.68 | 60.79 |
| DeepSeekMoE | HEAPr | 60M | 9.56 | 60.85 |
| | HEAPr | 120M | 9.47 | 60.89 |
| | DDP (Ours) | 30M | **9.38** | **61.84** |

We use LoRAP and HEAPr as representative baselines for dense and MoE models, respectively, and evaluate LoRA recovery under the same data and evaluation settings with different token budgets. This comparison examines whether post-pruning LoRA fine-tuning can compensate for a sub-optimal sparse structure, or whether directly optimizing the masks is more effective.

As shown in Table 7, LoRA recovery improves training-free pruned models as the token budget increases, but the gains remain limited and saturate below DDP. On LLaMA-2-7B, LoRAP improves from 59.44 to 62.98 mean accuracy with 120M recovery tokens, yet still trails DDP's 64.82 using only 30M mask-optimization tokens. Similarly, on DeepSeekMoE-16B, HEAPr gains only marginally even with 120M recovery tokens and remains worse than DDP in both perplexity and accuracy. These results suggest that the main bottleneck is the quality of the initial sparse structure: LoRA can adapt the remaining network after pruning, but cannot revise poor pruning decisions. In contrast, DDP directly optimizes the sparse support under the target budget while keeping pretrained weights frozen.

### 5.5. Optimal Sparsity Pattern

Our method automatically discovers interpretable structured sparsity patterns. For dense models, sparsity tends to concentrate in later layers, and attention shows higher cross-layer variance than MLPs, suggesting greater redundancy in multi-head attention. For MoE models, sparsity varies substantially across experts: rarely routed experts are pruned more aggressively, indicating a negative correlation between activation frequency and pruning ratio. The learned mask values are also diverse: they are centered around 1, but include values close to 0 and larger than 1, showing that DDP's expanded mask parameterization can both prune unimportant components and rescale retained ones. These trends align with prior heuristic observations and the conjecture that later layers contribute less to the output distribu-

*Table 8.* Additional comparison with Týr-the-Pruner on dense models under different sparsity ratios. We report WikiText-2 perplexity and mean zero-shot accuracy. Bold indicates the best result at the same model and sparsity ratio.

| Model | Method | 12.5% | | 25% | | 37.5% | | 50% | |
|---|---|---|---|---|---|---|---|---|---|
| | | Wiki PPL ↓ | Mean Acc. ↑ | Wiki PPL ↓ | Mean Acc. ↑ | Wiki PPL ↓ | Mean Acc. ↑ | Wiki PPL ↓ | Mean Acc. ↑ |
| LLaMA-2-7B | Týr-the-Pruner | 5.84 | 56.98 | 7.51 | 54.64 | 10.29 | 52.21 | 16.17 | 47.41 |
| | DDP (Ours) | **5.72** | **61.99** | **6.87** | **60.29** | **8.17** | **55.98** | **10.34** | **52.17** |
| LLaMA-2-13B | Týr-the-Pruner | **5.03** | 62.66 | 5.79 | 61.16 | 7.17 | 58.67 | 9.59 | 54.58 |
| | DDP (Ours) | **5.03** | **65.62** | **5.75** | **64.51** | **6.65** | **60.88** | **8.14** | **58.43** |
| Mistral-7B-v0.3 | Týr-the-Pruner | 5.61 | 63.05 | 7.08 | 60.22 | 10.25 | 52.34 | 15.53 | 46.21 |
| | DDP (Ours) | **5.59** | **66.31** | **6.73** | **62.85** | **7.80** | **58.39** | **9.88** | **50.87** |
| Mistral-Nemo | Týr-the-Pruner | 6.31 | 64.15 | 7.87 | 60.61 | 11.47 | 54.63 | 16.85 | 47.92 |
| | DDP (Ours) | **6.25** | **67.23** | **7.55** | **63.99** | **9.39** | **58.95** | **12.07** | **53.53** |

*Table 9.* Per-step training cost comparison on LLaMA2-7B with global batch size 16 and sequence length 2,048. Memory denotes the sum of activation memory, model memory, and gradient/optimizer memory. For LoRA, we use rank $r = 8$.

| Method | Params | Memory | FLOPs | Time / Step |
|---|---|---|---|---|
| DDP | 0.35M | 18 GB | 0.97E | 2.1 s |
| DDP (distill) | 0.35M | 19 GB | 1.254E | 2.5 s |
| LoRA | 20M | 18 GB | 0.97E | 2.4 s |
| Full FT | 7B | 83 GB | 1.40E | 2.9 s |

tion (Gupta et al., 2025), while MoE capacity is unevenly distributed across experts. Detailed results are provided in Section E.

## 5.6. Additional Results

Table 8 reports additional comparisons with Týr-the-Pruner (Li et al., 2025a) across more model families and sparsity levels. DDP consistently improves perplexity and downstream accuracy, with larger gains under higher sparsity. These results further demonstrate the robustness and applicability of DDP across model families and pruning ratios.

## 5.7. Computational Cost

We compare the per-step training cost of DDP against LoRA and full fine-tuning on LLaMA-7B. All methods use a global batch size of 16, a per-device batch size of 2 with gradient accumulation 2, sequence length 2,048, and activation checkpointing. We report trainable parameters, total memory, training FLOPs, and wall-clock time per step. As shown in Table 9, DDP achieves the lowest per-step time while updating only 0.35M mask parameters. The distillation variant adds only a modest overhead from the no-gradient teacher forward pass, without introducing additional trainable parameters or optimizer states. LoRA incurs extra cost from the low-rank adapter projections and their backward pass. As a result, DDP is faster than LoRA without distillation and remains comparable in wall-clock time with distillation, while using far fewer trainable parameters.

*Table 10.* End-to-end speedup results on dense models. We report Throughput (requests per second) and speedup ratio.

| Model | GPU | | 0% | 20% | 50% |
|---|---|---|---|---|---|
| LLaMA-7B | RTX 5090 | Through. | 10.88 | 14.75 | 23.98 |
| | | Speedup | 1.00× | 1.36x | 2.20x |
| LLaMA-13B | B200 | Through. | 27.20 | 32.14 | 38.56 |
| | | Speedup | 1.00× | 1.18x | 1.42x |

*Table 11.* End-to-end speedup results on MoE models. We report Throughput (requests per second) and speedup ratio.

| Model | GPU | | 0% | 20% | 40% | 60% |
|---|---|---|---|---|---|---|
| Qwen3 30B-A3B | B200 | Through. | 29.86 | 33.25 | 37.42 | 45.16 |
| | | Speedup | 1.00× | 1.11x | 1.25x | 1.51x |

## 5.8. Speedup

We measure end-to-end speedups with vLLM (Kwon et al., 2023) on 1,000 ShareGPT prompts and show the results in Tables 10 and 11. Speedups grow with sparsity and are more pronounced on memory-constrained devices. On RTX 5090, pruning LLaMA-7B achieves $1.36\times$ speedup at 20% sparsity and $2.20\times$ at 50% sparsity. On B200, gains are smaller but remain consistently positive. For MoE, Qwen3-30B-A3B on B200 scales well with expert pruning, reaching $1.11\times$, $1.25\times$, and $1.51\times$ speedup at 20%, 40%, and 60% sparsity, respectively. More implementation details are provided in Section C.

## 6. Conclusion

In this paper, we formulate structured pruning as an $\ell_0$-constrained mask optimization problem and propose a deterministic, mask-only approach that improves convergence stability and pruning quality with minimal compute. Across both dense and MoE LLMs, our method consistently outperforms prior pruning baselines by a large margin, and we further demonstrate practical deployability via end-to-end speedups. Future work will explore continued training to further close the accuracy gap under higher sparsity.

## Acknowledgements

The authors thank Siying Tao for assistance with figure preparation and Litong Deng for helpful discussions on the mathematical proofs. This work was supported by Ant Group, and we thank colleagues and collaborators at Ant Group for valuable discussions and technical support. This work was supported by the Fundamental and Interdisciplinary Disciplines Breakthrough Plan of the Ministry of Education of China (No. JYB2025XDXM101); the NSFC Projects (Nos. 62595773, 62376131, 62550004, 92270001). J.Z is also supported by the XPlorer Prize.

## Impact Statement

This paper presents work whose goal is to advance the field of Machine Learning. There are many potential societal consequences of our work, none of which we feel must be specifically highlighted here.

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

# A. Algorithm

---

**Algorithm 1** Deterministic Differentiable Pruning: Structured Pruning via Mask-Only Optimization

---

1: **Input:** target keep ratio $\rho$; total iterations $T$; number of heads $H$; MLP width $C$; number of layers $L$; prunable module types $\mathcal{N}$ (dense: {attn, mlp}; MoE: {expert}); distillation weight $\eta$
2: **Initialize:** for each $\tau \in \mathcal{N}$, initialize mask parameters $\boldsymbol{z}^\tau \leftarrow \boldsymbol{1}$ and multipliers $\lambda_1^\tau \leftarrow 0, \lambda_2^\tau \leftarrow 0, \lambda_3^\tau \leftarrow 0$.
3: **for** $t = 1, 2, \ldots, T$ **do**
4:  Compute progress $p_t \leftarrow t/T$ and update mapping sharpness $\mu_t \leftarrow \mu_0 - (\mu_0 - \mu_T)\sqrt{p_t}$
5:  **for** $\tau \in \mathcal{N}$ **do**
6:   Compute forward masks $\boldsymbol{m}^\tau \leftarrow \mathrm{ReLU}(\boldsymbol{z}^\tau)$
7:   Compute retention scores $\boldsymbol{s}^\tau$ from $\boldsymbol{z}^\tau$                (Eq. (12))
8:   Compute $\mathcal{L}_{\mathrm{sparsity}}^\tau$ and $\mathcal{L}_{\mathrm{bin}}^\tau$             (Eqs. (13), (15))
9:  **end for**
10:  Forward dense teacher and masked student to obtain output distributions $P_t$ and $P_s$ respectively, compute $\mathcal{L}_{\mathrm{kl}}$ (Eq. (17))
11:  Form $\mathcal{L}_{\mathrm{total}} = \mathcal{L}_{\mathrm{ce}}(\theta, \boldsymbol{m}) + \eta\mathcal{L}_{\mathrm{kl}}(\theta, \boldsymbol{m}) + \mathcal{L}_{\mathrm{sparsity}}(\boldsymbol{s}) + \mathcal{L}_{\mathrm{bin}}(\boldsymbol{s})$
12:  **for** $\tau \in \mathcal{N}$ **do**
13:   Update $\boldsymbol{z}^\tau$ by gradient descent on $\mathcal{L}_{\mathrm{total}}$.
14:   Update $\lambda_1^\tau, \lambda_2^\tau, \lambda_3^\tau$ by gradient ascent.
15:  **end for**
16: **end for**
17: **Pruning:** remove components whose corresponding mask value is 0 in $\boldsymbol{m}^\tau$ and merge nonzero mask values into model weights for each $\tau \in \mathcal{N}$, yielding the final pruned model.

---

# B. Theoretical Discussion

This section provides formal justification for the theoretical claims discussed in Section 3. We study *Deterministic Differentiable Pruning (DDP)* by (i) relating its relaxed keep budget constraint to the discrete $\ell_0$ active-set constraint, (ii) characterizing the effect of the binarization regularizer, and (iii) establishing a standard KKT-type convergence guarantee for deterministic augmented Lagrangian optimization on the (STE-induced) constrained surrogate. These ingredients are then combined to justify recovery of the exact hard $P$-budget in the annealed limit.

**Discrete $P$-budget formulation.** DDP parameterizes structured masks by logits $\boldsymbol{z} \in \mathbb{R}^K$ and applies a ReLU gate in the forward pass, $\boldsymbol{m}(\boldsymbol{z}) = \mathrm{ReLU}(\boldsymbol{z})$. Under this parameterization, enforcing an exact keep budget is equivalent to constraining the number of positive logits. We therefore define the discrete hard budget problem as

$$\min_{\boldsymbol{z}} \; J(\boldsymbol{z}) \quad \text{s.t.} \quad \sum_{k=1}^{K} \mathbb{I}[z_k > 0] = P, \tag{20}$$

where $P$ is the target number of active components. In terms of the keep ratio $\rho$, we set $P = \lceil \rho K \rceil$ (or $P = \rho K$ when $\rho K$ is an integer). Throughout, $J(z)$ can be any *deterministic* training objective; for example,

$$J(\boldsymbol{z}) \; = \; \mathcal{L}_{\mathrm{ce}}\big(\theta, \boldsymbol{m}(\boldsymbol{z})\big) \; + \; \eta\,\mathcal{L}_{\mathrm{kl}}\big(\theta, \boldsymbol{m}(\boldsymbol{z})\big),$$

with an optional distillation term $\mathcal{L}_{\mathrm{kl}}$.

**Relaxed keep budget.** To obtain a differentiable surrogate for the discrete active-set count, DDP uses the deterministic surrogate mapping $s_k = \phi(z_k; \mu) \in [0, 1]$ (cf. Equation (12)) and defines the relaxed keep budget

$$S_\mu(\boldsymbol{z}) = \sum_{k=1}^{K} \phi(z_k; \mu), \qquad c_\mu(\boldsymbol{z}) \triangleq S_\mu(\boldsymbol{z}) - P. \tag{21}$$

We emphasize that $S_\mu$ is used only for enforcing the budget constraint (via an augmented Lagrangian), while the forward mask is $\boldsymbol{m}(\boldsymbol{z}) = \mathrm{ReLU}(\boldsymbol{z})$.

## B.1. A basic property of the deterministic surrogate mapping

We start with a lightweight fact about $\phi(\cdot; \mu)$ that underlies the hard budget recovery argument: as $\mu \downarrow 0$, the relaxed keep budget $S_\mu(\boldsymbol{z})$ approaches the discrete count $\sum_k \mathbb{I}[z_k > 0]$ whenever the logits are separated from the ReLU threshold by a margin.

**Lemma B.1** (Margin-separated relaxation approaches the discrete count). *Assume $\phi(\cdot; \mu)$ is nondecreasing in $z$ for each $\mu > 0$ and satisfies pointwise convergence $\phi(z; \mu) \to \mathbb{I}[z > 0]$ as $\mu \downarrow 0$ for all $z \neq 0$ (as induced by Equation* (12)). *Fix $\delta > 0$ and define the active set $\mathcal{A}(\boldsymbol{z}) = \{k : z_k > 0\}$. Then there exists $\bar{\mu}(\delta) > 0$ such that for all $\mu \in (0, \bar{\mu}(\delta))$ and all $\boldsymbol{z}$ obeying the margin condition*

$$z_k \geq \delta \ \text{for } k \in \mathcal{A}(\boldsymbol{z}), \qquad z_k \leq -\delta \ \text{for } k \notin \mathcal{A}(\boldsymbol{z}),$$

*we have*

$$\big| S_\mu(\boldsymbol{z}) - |\mathcal{A}(\boldsymbol{z})| \big| \ \leq \ \tfrac{1}{2}.$$

*In particular, since both $S_\mu(\boldsymbol{z})$ and $|\mathcal{A}(\boldsymbol{z})|$ are integers whenever $S_\mu(\boldsymbol{z}) = P$, this implies $S_\mu(\boldsymbol{z}) = P \Rightarrow |\mathcal{A}(\boldsymbol{z})| = P$ for sufficiently small $\mu$ under the margin condition.*

*Proof.* By pointwise convergence, for the fixed $\delta > 0$ there exists $\bar{\mu}(\delta) > 0$ such that for all $\mu \in (0, \bar{\mu}(\delta))$,

$$\phi(\delta; \mu) \geq 1 - \frac{1}{4K}, \qquad \phi(-\delta; \mu) \leq \frac{1}{4K}.$$

By monotonicity, for $k \in \mathcal{A}(\boldsymbol{z})$ we have $\phi(z_k; \mu) \geq \phi(\delta; \mu)$, and for $k \notin \mathcal{A}(\boldsymbol{z})$ we have $\phi(z_k; \mu) \leq \phi(-\delta; \mu)$. Summing yields

$$S_\mu(\boldsymbol{z}) = \sum_{k \in \mathcal{A}(\boldsymbol{z})} \phi(z_k; \mu) + \sum_{k \notin \mathcal{A}(\boldsymbol{z})} \phi(z_k; \mu) \in \Big[ |\mathcal{A}(\boldsymbol{z})|\Big(1 - \tfrac{1}{4K}\Big), \ |\mathcal{A}(\boldsymbol{z})| + \tfrac{K - |\mathcal{A}(\boldsymbol{z})|}{4K} \Big],$$

hence $\big| S_\mu(\boldsymbol{z}) - |\mathcal{A}(\boldsymbol{z})| \big| \leq \tfrac{1}{2}$. $\qquad \square$

## B.2. Binarization loss

We next analyze the role of the binarization loss, which encourages the relaxed scores $\boldsymbol{s} = \phi(\boldsymbol{z}; \mu)$ to commit to $\{0, 1\}$ under the keep-budget constraint.

**Proposition B.2** (Binarization regularizer drives $\boldsymbol{s}$ toward $\{0, 1\}$). *Fix $\mu > 0$ and recall the relaxed keep budget $S_\mu(\boldsymbol{z}) = \sum_{k=1}^{K} \phi(z_k; \mu)$ and constraint $c_\mu(\boldsymbol{z}) = S_\mu(\boldsymbol{z}) - P$. Define*

$$B(\boldsymbol{z}) \triangleq \frac{1}{K} \sum_{k=1}^{K} s_k(1 - s_k), \qquad s_k = \phi(z_k; \mu) \in [0, 1],$$

*and the (scaled) binarization loss*

$$\mathcal{L}_{\text{bin}}(\boldsymbol{z}) \triangleq \lambda_3 B(\boldsymbol{z}).$$

*Consider the constrained problem*

$$\min_{\boldsymbol{z}} \ J(\boldsymbol{z}) + \mathcal{L}_{\text{bin}}(\boldsymbol{z}) \quad s.t. \quad c_\mu(\boldsymbol{z}) = 0, \tag{22}$$

*where $J(\boldsymbol{z})$ is any deterministic objective and $\boldsymbol{m}(\boldsymbol{z}) = \text{ReLU}(\boldsymbol{z})$ is the forward gate.*

*Assume: (i) J is bounded below on the feasible set $\{\boldsymbol{z} : c_\mu(\boldsymbol{z}) = 0\}$ by $J_{\text{inf}}$; and (ii) there exists a* binary feasible *point $\boldsymbol{z}^{\text{bin}}$ such that $c_\mu(\boldsymbol{z}^{\text{bin}}) = 0$ and $\boldsymbol{s}(\boldsymbol{z}^{\text{bin}}) \in \{0, 1\}^K$ (hence $B(\boldsymbol{z}^{\text{bin}}) = 0$ and $\mathcal{L}_{\text{bin}}(\boldsymbol{z}^{\text{bin}}) = 0$).*

*Let $\boldsymbol{z}_{\lambda_3}$ be any global minimizer of* (22). *Then*

$$B(\boldsymbol{z}_{\lambda_3}) \leq \frac{J(\boldsymbol{z}^{\text{bin}}) - J_{\text{inf}}}{\lambda_3}, \tag{23}$$

*and the average distance of $\boldsymbol{s}$ to the binary set is bounded by*

$$\frac{1}{K} \sum_{k=1}^{K} \min\{s_k, \ 1 - s_k\} \ \leq \ 2\,B(\boldsymbol{z}_{\lambda_3}) \ \leq \ \frac{2\big(J(\boldsymbol{z}^{\text{bin}}) - J_{\text{inf}}\big)}{\lambda_3}. \tag{24}$$

*Consequently, as $\lambda_3 \to \infty$, any sequence of minimizers $\{z_{\lambda_3}\}$ satisfies $B(z_{\lambda_3}) \to 0$, and the relaxed scores $s$ become (on average) arbitrarily close to $\{0, 1\}^K$.*

*Proof.* Let $z_{\lambda_3}$ be a global minimizer of (22), and let $z^{\mathrm{bin}}$ be the binary feasible point. Since both are feasible and $\mathcal{L}_{\mathrm{bin}}(z^{\mathrm{bin}}) = 0$, optimality implies

$$J(z_{\lambda_3}) + \lambda_3 B(z_{\lambda_3}) \;\leq\; J(z^{\mathrm{bin}}) + \lambda_3 B(z^{\mathrm{bin}}) = J(z^{\mathrm{bin}}).$$

Rearranging gives

$$\lambda_3 B(z_{\lambda_3}) \leq J(z^{\mathrm{bin}}) - J(z_{\lambda_3}) \leq J(z^{\mathrm{bin}}) - J_{\mathrm{inf}},$$

where the last inequality uses $J(z_{\lambda_3}) \geq J_{\mathrm{inf}}$ on the feasible set. Dividing by $\lambda_3$ yields (23).

For (24), note that for any $s \in [0, 1]$,

$$\min\{s, 1 - s\} \leq 2s(1 - s).$$

Applying this elementwise and averaging gives

$$\frac{1}{K} \sum_{k=1}^{K} \min\{s_k, 1 - s_k\} \leq \frac{1}{K} \sum_{k=1}^{K} 2s_k(1 - s_k) = 2B(z_{\lambda_3}),$$

and the final bound follows from (23). $\qquad\square$

## B.3. Convergence of deterministic augmented Lagrangian optimization

For any fixed $\mu > 0$, the equality-constrained surrogate induced by the deterministic mapping $\phi(\cdot\,; \mu)$ can be optimized with an inexact augmented Lagrangian / method-of-multipliers scheme. The following theorem states a standard KKT accumulation-point guarantee for the deterministic iterates, under an STE-surrogate interpretation.

**Theorem B.3** (Deterministic inexact ALM yields KKT accumulation points (STE surrogate)). *Fix an annealing level $\mu > 0$ and a binarization weight $\lambda_3 \geq 0$, and define the relaxed keep budget*

$$S_\mu(z) \triangleq \sum_{k=1}^{K} \phi(z_k; \mu), \qquad c(z) \triangleq S_\mu(z) - P,$$

*where $\phi(\cdot\,; \mu)$ is the deterministic surrogate mapping in Equation (12) and $P$ is the target number of active components. Let the forward gate be $m(z) = \mathrm{ReLU}(z)$ as in Equation (11). Define the relaxed-score vector $s(z) \in [0, 1]^K$ by $s_k(z) = \phi(z_k; \mu)$, and*

$$B(z) \triangleq \frac{1}{K} \sum_{k=1}^{K} s_k(z)\big(1 - s_k(z)\big), \qquad \mathcal{L}_{\mathrm{bin}}(z) \triangleq \lambda_3 \, B(z).$$

*Let*

$$F(z) \triangleq J(z) + \mathcal{L}_{\mathrm{bin}}(z),$$

*where $J(z)$ can be any deterministic training loss (e.g., $\mathcal{L}_{\mathrm{ce}}(\theta, m(z))$ plus optional distillation terms). Consider the equality-constrained STE-induced surrogate problem*

$$\min_z \; F(z) \quad s.t. \quad c(z) = 0. \tag{25}$$

*Let $\{z_t, \nu_t, \gamma_t\}$ be generated by an inexact method-of-multipliers (augmented Lagrangian) scheme applied to*

$$\mathcal{L}_{\gamma_t}(z, \nu) \triangleq F(z) + \nu \, c(z) + \frac{\gamma_t}{2} \, c(z)^2,$$

*i.e., each outer iteration performs (possibly inexact) descent steps on $z$ and then updates*

$$\nu_{t+1} = \nu_t + \gamma_t \, c(z_t),$$

*with $\gamma_t$ nondecreasing (and increased when feasibility stalls).*

*Assume:*

*(A1)* (***STE-surrogate regularity***) *$F$ and $c$ are continuously differentiable on a neighborhood containing all iterates, where $\nabla F$ and $\nabla c$ are interpreted as the gradients induced by the chosen STE for* ReLU/Clamp.

*(A2)* (***Constraint nondegeneracy on iterates***) *There exists $\underline{\gamma} > 0$ such that $\|\nabla c(z_t)\| \geq \underline{\gamma}$ for all sufficiently large $t$. (In particular, this implies LICQ at feasible limit points.)*

*(A3)* (***Inexact primal stationarity***) *The inexact primal update satisfies*

$$\left\| \nabla_z \mathcal{L}_{\gamma_t}(z_t, \nu_t) \right\| \leq \varepsilon_t, \qquad \text{with } \varepsilon_t \to 0.$$

*(A4)* (***Bounded iterates and penalty growth***) *The iterates $\{z_t\}$ remain in a compact set, $\{\gamma_t\}$ is nondecreasing, and $\gamma_t$ is increased without bound if feasibility stalls (i.e., if $|c(z_t)|$ does not decrease sufficiently).*

*Then any accumulation point $z^\star$ of $\{z_t\}$ is feasible ($c(z^\star) = 0$), and there exists $\nu^\star$ such that $(z^\star, \nu^\star)$ satisfies the first-order KKT conditions of* (25):

$$\nabla F(z^\star) + \nu^\star \nabla c(z^\star) = 0, \qquad c(z^\star) = 0.$$

*Proof.* Fix $\mu > 0$ and $\lambda_3 \geq 0$, and write $c(z) = S_\mu(z) - P$. Define the effective multiplier

$$\tilde{\nu}_t \triangleq \nu_t + \gamma_t \, c(z_t).$$

By the multiplier update, $\tilde{\nu}_t = \nu_{t+1}$. The inexact primal condition (A3) expands as

$$\left\| \nabla F(z_t) + \tilde{\nu}_t \, \nabla c(z_t) \right\| = \left\| \nabla_z \mathcal{L}_{\gamma_t}(z_t, \nu_t) \right\| \leq \varepsilon_t. \tag{26}$$

**Step 1: Boundedness of gradients and $\tilde{\nu}_t$.** By (A4), $\{z_t\}$ lies in some compact set $\mathcal{Z}$. By (A1), $\nabla F$ and $\nabla c$ are continuous on $\mathcal{Z}$ and thus bounded: there exists $G < \infty$ such that $\|\nabla F(z)\| \leq G$ for all $z \in \mathcal{Z}$. Combining (26) with the triangle inequality yields

$$|\tilde{\nu}_t| \, \|\nabla c(z_t)\| \leq \|\nabla F(z_t)\| + \varepsilon_t \leq G + \varepsilon_t.$$

Using (A2), for all sufficiently large $t$ we have $\|\nabla c(z_t)\| \geq \underline{\gamma}$, hence

$$|\tilde{\nu}_t| \leq \frac{G + \varepsilon_t}{\underline{\gamma}}, \tag{27}$$

so $\{\tilde{\nu}_t\}$ (equivalently $\{\nu_{t+1}\}$) is bounded for all large $t$.

**Step 2: Feasibility of accumulation points.** Let $z^\star$ be any accumulation point of $\{z_t\}$, and suppose for contradiction that $c(z^\star) \neq 0$. Then there exists a subsequence $t_j$ such that $z_{t_j} \to z^\star$, and by continuity of $c$, $|c(z_{t_j})| \to |c(z^\star)| > 0$, i.e., $|c(z_{t_j})|$ is bounded away from 0 for large $j$. Under (A4), persistent nonvanishing constraint violation triggers $\gamma_{t_j} \to \infty$ along such a subsequence. But then

$$\tilde{\nu}_{t_j} = \nu_{t_j} + \gamma_{t_j} c(z_{t_j}) = \nu_{t_j+1}$$

must diverge in magnitude because $\gamma_{t_j}|c(z_{t_j})| \to \infty$, contradicting the boundedness of $\{\tilde{\nu}_t\}$ from (27). Therefore, every accumulation point must satisfy $c(z^\star) = 0$.

**Step 3: KKT stationarity at accumulation points.** Take a convergent subsequence $z_{t_j} \to z^\star$. By Step 2, $c(z^\star) = 0$. By Step 1, $\{\nu_{t_j+1}\} = \{\tilde{\nu}_{t_j}\}$ is bounded and hence admits a convergent sub-subsequence; relabel so that $\nu_{t_j+1} \to \nu^\star$. From (26),

$$\left\| \nabla F(z_{t_j}) + \nu_{t_j+1} \nabla c(z_{t_j}) \right\| \leq \varepsilon_{t_j} \to 0.$$

Using continuity of $\nabla F$ and $\nabla c$ (A1), letting $j \to \infty$ yields $\nabla F(z^\star) + \nu^\star \nabla c(z^\star) = 0$. Together with feasibility $c(z^\star) = 0$, this is exactly the first-order KKT condition for (25). $\qquad\square$

**Remark (iterates need not be feasible).** The primal iterates $z_t$ produced by the (inexact) descent steps generally do *not* satisfy $c(z_t) = 0$. Instead, the augmented penalty term $\frac{\gamma_t}{2} c(z)^2$ and multiplier update drive feasibility over outer iterations, and the theorem guarantees feasibility only at accumulation points.

## B.4. Final theorem

We now combine Lemma B.1 (relaxed budget approaches the discrete count under a margin), Proposition B.2 (binarization drives $s$ toward $\{0,1\}^K$), and Theorem B.3 (deterministic ALM admits KKT accumulation points at each fixed $\mu$), to justify that staged DDP recovers the *exact* hard $P$-budget for sufficiently small annealing levels.

**Theorem B.4** (DDP recovers the exact hard $P$-budget in the annealed limit). *Let $m(z) = \mathrm{ReLU}(z)$ and define the discrete hard-budget problem*

$$\min_{z} \ J(z) \quad s.t. \quad \sum_{k=1}^{K} \mathbb{I}[z_k > 0] = P, \tag{28}$$

*where $J(z)$ is any deterministic training objective (e.g., $\mathcal{L}_{\mathrm{ce}}(\theta, m(z))$ plus optional distillation terms) and $P = [\rho K]$.*

*For each $\mu > 0$, define the relaxed keep budget and constraint*

$$S_\mu(z) \triangleq \sum_{k=1}^{K} \phi(z_k; \mu), \qquad c_\mu(z) \triangleq S_\mu(z) - P,$$

*where $\phi(\cdot; \mu)$ is the deterministic surrogate mapping in Equation (12). Define the relaxed-score vector $s_\mu(z) \in [0,1]^K$ by $s_{\mu,k}(z) = \phi(z_k; \mu)$, and the (unscaled) binarization measure*

$$B_\mu(z) \triangleq \frac{1}{K} \sum_{k=1}^{K} s_{\mu,k}(z)\big(1 - s_{\mu,k}(z)\big), \qquad \mathcal{L}_{\mathrm{bin}}(z) \triangleq \lambda_3 \, B_\mu(z).$$

*Consider the relaxed constrained surrogate at annealing level $\mu$:*

$$\min_{z} \ F_{\mu,\lambda_3}(z) \triangleq J(z) + \mathcal{L}_{\mathrm{bin}}(z) \quad s.t. \quad c_\mu(z) = 0. \tag{29}$$

*Run DDP in outer rounds $r = 1, 2, \ldots$ with $\mu_r \downarrow 0$ and $\lambda_{3,r} \uparrow \infty$. At round $r$, apply an inexact method-of-multipliers (augmented Lagrangian) scheme (with STE-induced gradients for $\mathrm{ReLU}/\mathrm{Clamp}$) to (29) using*

$$\mathcal{L}_\gamma(z, \nu) = F_{\mu_r, \lambda_{3,r}}(z) + \nu \, c_{\mu_r}(z) + \frac{\gamma}{2} c_{\mu_r}(z)^2, \qquad \nu \leftarrow \nu + \gamma \, c_{\mu_r}(z).$$

*Assume the following hold for each round $r$:*

*(A1)* (**Regularity of the STE surrogate**) *Under the STE-induced surrogate interpretation, $F_{\mu_r, \lambda_{3,r}}$ and $c_{\mu_r}$ are $C^1$ on a neighborhood containing the iterates, and $\nabla c_{\mu_r}(z^\star) \neq 0$ at any feasible accumulation point $z^\star$ (LICQ-type nondegeneracy).*

*(A2)* (**KKT accumulation point at round** $r$) *The round-$r$ iterates remain bounded and admit an accumulation point $z^{(r)}$ such that*

$$c_{\mu_r}(z^{(r)}) = 0, \qquad \exists \nu^{(r)} : \ \nabla F_{\mu_r, \lambda_{3,r}}(z^{(r)}) + \nu^{(r)} \nabla c_{\mu_r}(z^{(r)}) = 0.$$

*(A3)* (**Binarization**) *$B_{\mu_r}(z^{(r)}) \to 0$ as $r \to \infty$ (equivalently, $\mathcal{L}_{\mathrm{bin}}(z^{(r)})/\lambda_{3,r} \to 0$), e.g., as ensured by Proposition B.2 under its conditions.*

*(A4)* (**ReLU-stable margin**) *There exists $\delta > 0$ and $r_0$ such that for all $r \geq r_0$,*

$$z_k^{(r)} \geq \delta \text{ for } k \in \mathcal{A}^{(r)} \triangleq \{k : z_k^{(r)} > 0\}, \qquad z_k^{(r)} \leq -\delta \text{ for } k \notin \mathcal{A}^{(r)}.$$

*Then there exists $r_1$ such that for all $r \geq r_1$, the discrete mask $b^{(r)} \in \{0,1\}^K$ induced by the forward gate,*

$$b_k^{(r)} \triangleq \mathbb{I}[m_k(z_k^{(r)}) > 0] = \mathbb{I}[z_k^{(r)} > 0],$$

*satisfies the* exact *hard budget:*

$$\sum_{k=1}^{K} b_k^{(r)} = P.$$

*Proof of Theorem B.4.* Fix any round $r$. By (A2), $\boldsymbol{z}^{(r)}$ is feasible for the relaxed constraint, hence $S_{\mu_r}(\boldsymbol{z}^{(r)}) = P$.

Let $\mathcal{A}^{(r)} = \{k : z_k^{(r)} > 0\}$, so the hard-budget count equals $\sum_{k=1}^{K} \mathbb{I}[z_k^{(r)} > 0] = |\mathcal{A}^{(r)}|$. By the margin condition (A4), $z_k^{(r)} \geq \delta$ for $k \in \mathcal{A}^{(r)}$ and $z_k^{(r)} \leq -\delta$ otherwise. Since $\phi(\cdot; \mu)$ is monotone and converges pointwise to $\mathbb{I}[z > 0]$ as $\mu \downarrow 0$ (by the construction in Equation (12)), there exists $\bar{\mu}(\delta) > 0$ such that for all $\mu \in (0, \bar{\mu}(\delta))$,

$$\phi(\delta; \mu) \geq 1 - \frac{1}{4K}, \qquad \phi(-\delta; \mu) \leq \frac{1}{4K}.$$

Choose $r_1$ such that $\mu_r < \bar{\mu}(\delta)$ for all $r \geq r_1$.

For such $r$, using monotonicity of $\phi(\cdot; \mu_r)$,

$$\sum_{k \in \mathcal{A}^{(r)}} \phi(z_k^{(r)}; \mu_r) \geq |\mathcal{A}^{(r)}|\Big(1 - \frac{1}{4K}\Big), \qquad \sum_{k \notin \mathcal{A}^{(r)}} \phi(z_k^{(r)}; \mu_r) \leq (K - |\mathcal{A}^{(r)}|)\frac{1}{4K}.$$

Hence

$$\Big| S_{\mu_r}(\boldsymbol{z}^{(r)}) - |\mathcal{A}^{(r)}| \Big| = \Big| \sum_{k=1}^{K} \phi(z_k^{(r)}; \mu_r) - \sum_{k=1}^{K} \mathbb{I}[z_k^{(r)} > 0] \Big| \leq \frac{1}{2}.$$

But $S_{\mu_r}(\boldsymbol{z}^{(r)}) = P$ and $|\mathcal{A}^{(r)}|$ are integers, so the only possibility is $|\mathcal{A}^{(r)}| = P$, i.e., $\sum_{k=1}^{K} \mathbb{I}[z_k^{(r)} > 0] = P$. Finally, (A3) (e.g., Proposition B.2) provides a sufficient condition under which increasing $\lambda_{3,r}$ drives the *unscaled* binarization measure $B_{\mu_r}(\boldsymbol{z}^{(r)})$ toward 0, which empirically promotes the margin separation (A4) by discouraging fractional relaxed scores. $\square$

## C. Additional Implementation Details

In this section, we report the hyperparameters used in our experiments. Unless otherwise noted, we optimize mask parameters with AdamW ($\beta_1$=0.9, $\beta_2$=0.999) using a batch size of 16, 1,000 training steps, and a context length of 2,048. We use $\mu_T = 0.05$ and apply a linear warmup stage followed by a cosine decay schedule for the learning rate. We search for the distillation coefficient $\eta$ and use $\eta$=2 for all models with 20% sparsity in the main results. We select a learning rate of $2 \times 10^{-2}$ for all latent variable $\boldsymbol{z}$ as well as $\lambda_1$ and $\lambda_3$; for $\lambda_2$ we select a learning rate of $4 \times 10^{-1}$. For higher sparsity ratio, we used a higher learning rate of $8 \times 10^{-1}$ for $\lambda_2$.

For the vLLM speedup experiments, we follow the default settings in the official repository and evaluate our pruned models on 1,000 real prompts sampled from ShareGPT. We use the default scheduler and measure end-to-end wall-clock latency/throughput on same GPU for both dense and pruned models.

## D. Full Results

This section provides detailed results supporting the main text and additional pruning results on other model families, including dense Qwen3 models (Tables 12 to 14). Across the Qwen3 dense series, we observe a clear scaling trend: pruning is consistently less lossy for larger models. For example, at 20% pruning, Qwen3-32B only drops slightly in average zero-shot accuracy (72.12→71.09), while Qwen3-4B degrades more noticeably (65.38→62.99). At 50% pruning, the gap widens further (32B: 66.48 vs. 4B: 47.63), suggesting substantially higher redundancy and robustness at larger scale.

*Table 12.* Detailed performance of pruned dense LLMs under different pruning methods. All pruning baseline methods are evaluated after fine-tuning. **Bold** indicates the best performance at the same pruning ratio.

| Ratio | Method | Wiki2 | BoolQ | PIQA | HellaSwag | WinoGrande | ARC-e | ARC-c | OBQA | Average |
|---|---|---|---|---|---|---|---|---|---|---|
| \multicolumn{11}{c}{LLaMA-7B} |
| Dense | Original | 12.62 | 76.50 | 79.35 | 76.00 | 69.53 | 72.10 | 44.28 | 44.00 | 65.96 |
| 20% | LoRAPrune | 16.80 | 65.62 | 79.31 | 70.00 | 62.76 | 65.87 | 37.69 | 39.14 | 60.05 |
| | LoRAP | 16.35 | 72.94 | 76.93 | 70.90 | 65.75 | 64.31 | 39.93 | 41.20 | 61.70 |
| | SlimLLM | 15.55 | **74.71** | 76.61 | 71.23 | 66.54 | 66.96 | 40.61 | 40.20 | 62.41 |
| | Ours | **15.20** | 74.10 | **77.97** | **73.20** | **69.22** | **69.65** | **42.41** | **42.40** | **64.13** |
| 50% | LoRAPrune | 30.12 | 61.88 | 71.53 | 47.86 | 55.01 | 45.13 | 31.62 | 34.98 | 49.71 |
| | LoRAP | 30.90 | 63.00 | 69.64 | 54.42 | 58.41 | 51.94 | 32.00 | 35.80 | 52.17 |
| | SlimLLM | 26.71 | 62.78 | 68.99 | 54.73 | 61.01 | 54.55 | 33.28 | 36.80 | 53.16 |
| | Ours | **26.70** | **67.77** | **71.49** | **60.36** | **62.67** | **58.54** | **35.49** | 36.20 | **56.07** |
| \multicolumn{11}{c}{LLaMA-2-7B} |
| Dense | Original | 12.18 | 79.30 | 78.29 | 76.11 | 69.85 | 73.86 | 44.80 | 44.20 | 66.63 |
| 20% | LoRAP | 14.67 | 70.89 | **78.13** | 69.93 | 65.67 | 65.99 | 38.48 | 39.60 | 61.2 |
| | SlimLLM | 15.28 | 72.29 | 78.02 | 70.95 | 64.88 | 67.17 | 38.99 | 39.60 | 61.70 |
| | Ours | **14.39** | **75.11** | 77.31 | **73.70** | **68.90** | **72.35** | **43.00** | **43.40** | **64.82** |
| 50% | LoRAP | **26.26** | 63.27 | 70.78 | 55.14 | 57.85 | 52.15 | 30.97 | 36.00 | 52.31 |
| | SlimLLM | 27.29 | 64.19 | 69.04 | 53.60 | 55.33 | 52.53 | 32.08 | 37.40 | 52.02 |
| | Ours | 26.34 | **66.27** | **72.42** | **60.39** | **62.59** | **61.03** | **36.60** | **37.60** | **56.70** |
| \multicolumn{11}{c}{LLaMA-13B} |
| Dense | Original | 11.58 | 79.88 | 79.87 | 79.33 | 73.48 | 73.23 | 48.38 | 47.40 | 68.79 |
| 20% | LoRAP | 13.58 | 73.39 | 78.73 | 75.54 | 69.30 | **70.62** | 43.00 | 42.40 | 64.71 |
| | SlimLLM | 13.35 | 74.13 | 77.53 | 74.73 | 69.30 | 70.45 | 42.32 | 41.00 | 64.21 |
| | Ours | **12.71** | **77.00** | **78.94** | **77.14** | **71.98** | 70.29 | **45.90** | **47.00** | **66.89** |
| 50% | LoRAP | 22.66 | 72.29 | 74.10 | 63.29 | 62.83 | 60.82 | 35.24 | 38.80 | 58.20 |
| | SlimLLM | 25.64 | 62.31 | 72.20 | 60.84 | 60.62 | 55.60 | 33.87 | 37.80 | 54.75 |
| | Ours | **20.32** | **75.08** | **75.30** | **69.74** | **69.85** | **64.90** | **39.51** | **40.60** | **62.14** |

# E. Analysis

We visualize learned sparsity patterns across model families. For dense models, we plot (i) attention-head sparsity as a layer-by-head heatmap and (ii) layer-wise MLP sparsity (number of zeroed channels per layer). Results for LLaMA-7B at 20% and 50% sparsity are shown in Figures 4 and 5. For MoE models, we plot expert sparsity as a layer-by-expert heatmap in Figure 6.

**Dense models (LLaMA-7B).** At 20% sparsity, head pruning is conservative, with most heads retained and sparsity concentrated in a small subset of layers/heads (Figure 4b). At 50% sparsity, the head map becomes markedly more selective and structured (Figure 5b), indicating substantial redundancy in multi-head attention and a tendency to concentrate capacity into fewer effective heads. For MLP channels, we observe a "U-shaped" layer-wise trend: higher sparsity in the first layers, a denser early-to-mid region, and increasing sparsity toward later layers with a slight drop at the end (Figures 4a and 5a).

**MoE models (DeepSeekMoE-16B).** Expert sparsity is highly non-uniform: even at 20% sparsity, pruning concentrates on consistently under-utilized experts (Figure 6a). As sparsity increases (40%–60%), the model preserves a core set of frequently routed experts while pruning rarely selected ones more aggressively (Figures 6b and 6c). This imbalance helps explain MoE robustness under pruning, since large savings can be obtained by removing low-utility experts with limited impact on the dominant computation paths.

*Table 13.* Detailed performance of pruned Qwen3 dense LLMs using our method.

| Ratio | Method | Wiki2 | BoolQ | OBQA | RTE | WinoGrande | HellaSwag | PIQA | MathQA | ARC-e | ARC-c | Average |
|---|---|---|---|---|---|---|---|---|---|---|---|---|
| **Qwen3-32B** | | | | | | | | | | | | |
| Dense | Original | 7.61 | 86.33 | 46.20 | 76.53 | 72.93 | 82.57 | 82.15 | 58.32 | 83.21 | 60.84 | 72.12 |
| 20% | Ours | 7.25 | 81.77 | 46.80 | 76.17 | 75.37 | 80.44 | 80.58 | 52.26 | 83.63 | 62.80 | 71.09 |
| 50% | Ours | 9.78 | 82.81 | 44.80 | 74.37 | 71.19 | 73.31 | 77.80 | 37.76 | 80.89 | 55.38 | 66.48 |
| **Qwen3-14B** | | | | | | | | | | | | |
| Dense | Original | 8.65 | 89.30 | 46.20 | 77.62 | 72.45 | 78.82 | 80.03 | 56.85 | 82.87 | 60.32 | 71.61 |
| 20% | Ours | 8.78 | 88.01 | 46.20 | 77.26 | 73.95 | 77.26 | 79.76 | 50.32 | 82.91 | 60.32 | 70.67 |
| 50% | Ours | 14.24 | 76.09 | 41.40 | 60.65 | 63.77 | 64.14 | 74.54 | 28.44 | 71.68 | 45.73 | 58.49 |
| **Qwen3-8B** | | | | | | | | | | | | |
| Dense | Original | 9.72 | 86.51 | 41.40 | 77.62 | 68.03 | 74.93 | 77.48 | 49.45 | 80.68 | 56.23 | 68.04 |
| 20% | Ours | 9.75 | 84.16 | 39.60 | 75.45 | 68.90 | 72.77 | 77.53 | 42.81 | 77.69 | 54.10 | 65.89 |
| 50% | Ours | 17.31 | 66.02 | 34.80 | 55.23 | 59.67 | 55.64 | 69.75 | 24.32 | 63.72 | 37.20 | 51.82 |
| **Qwen3-4B** | | | | | | | | | | | | |
| Dense | Original | 13.67 | 85.08 | 40.00 | 75.81 | 65.90 | 68.32 | 74.76 | 46.50 | 78.20 | 53.84 | 65.38 |
| 20% | Ours | 11.71 | 82.08 | 40.80 | 71.48 | 65.67 | 66.17 | 75.84 | 38.36 | 76.94 | 49.57 | 62.99 |
| 50% | Ours | 20.16 | 51.44 | 33.40 | 53.79 | 56.67 | 49.58 | 69.64 | 23.28 | 58.21 | 32.68 | 47.63 |

**Magnitude Distribution of Learned Masks.** We visualize the final learned mask values for Qwen3-8B head and intermediate masks, as well as Qwen3-30B-A3B expert masks, in Figure 7. Across all cases, the distributions are centered near 1, indicating that most retained components preserve their original scale. At the same time, the masks exhibit nontrivial variation, including values close to 0 and occasional values larger than 1, especially for intermediate and expert masks. This confirms that DDP does not merely learn binary keep-or-prune decisions, but can also rescale retained components through its expanded mask parameterization. The broader distributions for intermediate and expert masks further suggest greater flexibility in redistributing capacity across fine-grained MLP channels and MoE experts.

*Table 14.* Detailed performance of compressed Mixture of Experts LLMs under different pruning methods. **Bold** indicates the best performance at the same pruning ratio.

| Ratio | Method | C4 | BoolQ | OBQA | RTE | WinoGrande | HellaSwag | PIQA | MathQA | ARC-e | ARC-c | Average |
|---|---|---|---|---|---|---|---|---|---|---|---|---|
| | | | | | | **DeepSeekMoE-16B-Base** | | | | | | |
| Dense | Original | 9.05 | 72.45 | 44.00 | 63.54 | 70.24 | 77.37 | 80.68 | 31.52 | 72.89 | 47.86 | 62.28 |
| 20% | NAEE | 10.07 | 67.83 | 42.40 | 62.09 | 69.53 | 74.63 | 78.34 | 30.99 | 72.56 | 46.24 | 60.51 |
| | $D^2$-MoE | 12.62 | 69.32 | 41.40 | 61.01 | 69.22 | 69.87 | 76.44 | 29.45 | 71.29 | 42.75 | 58.97 |
| | Camera-P | 9.84 | 68.01 | 44.00 | **64.62** | 70.17 | 75.02 | 78.62 | 31.46 | 71.80 | 45.56 | 61.03 |
| | HEAPr | 9.85 | 64.77 | **44.60** | 56.68 | 70.72 | 76.59 | 79.87 | 31.83 | **73.91** | 46.59 | 60.62 |
| | Ours | **9.38** | **74.10** | 43.00 | 59.93 | **71.11** | **77.19** | **80.14** | **32.03** | 72.18 | **46.93** | **61.84** |
| 40% | NAEE | 12.80 | 62.26 | 39.60 | 57.40 | 63.69 | 66.16 | 75.41 | 27.37 | 64.06 | 38.48 | 54.94 |
| | $D^2$-MoE | 17.22 | 66.05 | 36.60 | 57.03 | 66.77 | 58.74 | 71.44 | 27.67 | 66.03 | 38.57 | 54.32 |
| | Camera-P | 11.68 | 70.64 | **43.20** | **58.48** | 68.51 | 69.04 | 75.41 | 29.01 | 70.71 | 42.24 | 58.58 |
| | HEAPr | 12.34 | 62.05 | 41.20 | 52.71 | 68.59 | 70.72 | 76.88 | 30.18 | **72.10** | 44.88 | 57.70 |
| | Ours | **10.75** | **73.64** | 42.40 | 56.32 | **70.48** | **74.93** | **79.05** | 30.99 | 71.17 | **44.45** | **60.38** |
| 60% | NAEE | 29.44 | 51.65 | 30.60 | 58.48 | 53.67 | 47.50 | 65.88 | 22.31 | 48.90 | 28.50 | 45.28 |
| | $D^2$-MoE | 34.54 | 61.78 | 31.60 | 53.43 | 61.09 | 43.29 | 63.87 | 23.65 | 50.59 | 31.14 | 46.72 |
| | Camera-P | 18.10 | 62.60 | 40.20 | 56.32 | 64.33 | 56.53 | 67.90 | 26.16 | 54.88 | 35.67 | 51.62 |
| | HEAPr | 21.25 | 61.83 | 39.40 | 53.79 | 63.22 | 53.70 | 69.97 | 26.43 | 61.07 | 33.02 | 51.38 |
| | Ours | **12.65** | **71.53** | **42.40** | **55.60** | **67.48** | **71.15** | **78.02** | **27.40** | **67.89** | **42.15** | **58.18** |
| | | | | | | **Qwen3-30B-A3B** | | | | | | |
| Dense | Original | 12.13 | 88.81 | 45.20 | 82.67 | 70.32 | 77.68 | 80.79 | 58.96 | 79.17 | 56.14 | 71.08 |
| 20% | NAEE | 12.44 | **88.74** | 44.40 | **83.39** | 69.85 | 77.32 | 80.14 | 49.27 | 77.31 | **56.31** | 69.64 |
| | $D^2$-MoE | 18.52 | 85.90 | 42.80 | 80.87 | 68.19 | 74.26 | 78.40 | 48.81 | 71.46 | 46.50 | 66.35 |
| | Camera-P | 12.25 | 88.50 | 44.40 | 82.67 | 70.56 | 78.51 | 80.30 | **52.80** | 78.33 | 54.43 | 69.94 |
| | Ours | **11.58** | 87.61 | **46.60** | 78.98 | **71.67** | **78.16** | **80.69** | 51.66 | **78.91** | 56.14 | **70.04** |
| 40% | NAEE | 13.87 | 87.21 | 43.00 | 74.01 | 68.74 | 73.50 | 77.74 | 42.91 | 72.72 | 52.13 | 65.77 |
| | $D^2$-MoE | 35.48 | 85.50 | 42.00 | 76.53 | 64.33 | 69.87 | 71.44 | 40.90 | 69.78 | 44.62 | 62.77 |
| | Camera-P | 15.58 | **86.97** | 43.00 | **79.42** | 67.64 | 73.79 | 78.51 | **44.96** | 77.31 | **55.55** | 67.46 |
| | Ours | **12.10** | 85.20 | **44.00** | 78.70 | **72.77** | **77.22** | **80.09** | 39.40 | **77.35** | 54.27 | **67.65** |
| 60% | NAEE | 19.37 | 72.23 | 36.60 | 68.95 | 63.93 | 62.06 | 71.87 | 28.68 | 66.92 | 42.92 | 57.13 |
| | $D^2$-MoE | 68.36 | 70.64 | 35.40 | 64.62 | 59.12 | 58.21 | 63.23 | 26.00 | 57.37 | 33.57 | 52.02 |
| | Camera-P | 24.48 | **82.08** | 41.00 | 68.95 | 64.64 | 64.90 | 73.01 | 30.89 | 62.46 | 43.35 | 59.03 |
| | Ours | **13.54** | 80.34 | **43.80** | **69.68** | **70.17** | **73.36** | **77.04** | **31.66** | **73.70** | **50.43** | **63.35** |

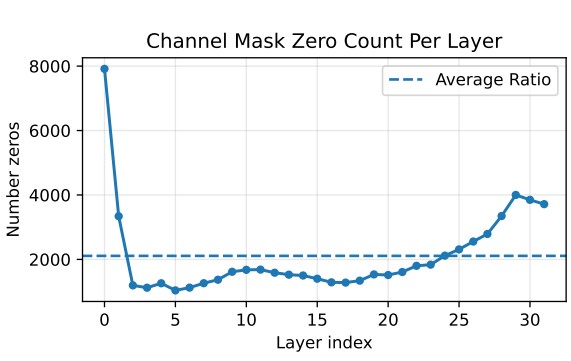

*(a)* MLP channel sparsity (per layer).

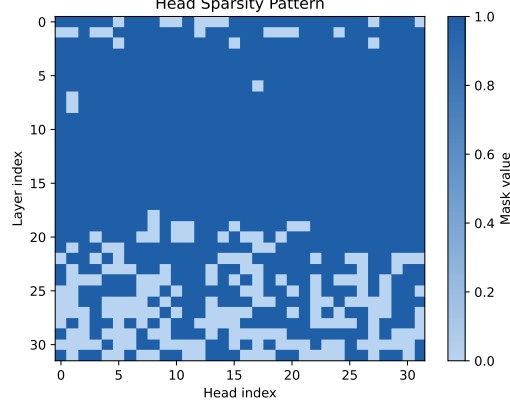

*(b)* Attention-head sparsity (layer×head).

*Figure 4.* **Dense-model sparsity patterns (LLaMA-7B, 20% sparsity).** Left: layer-wise MLP channel sparsity. Right: learned head sparsity map.

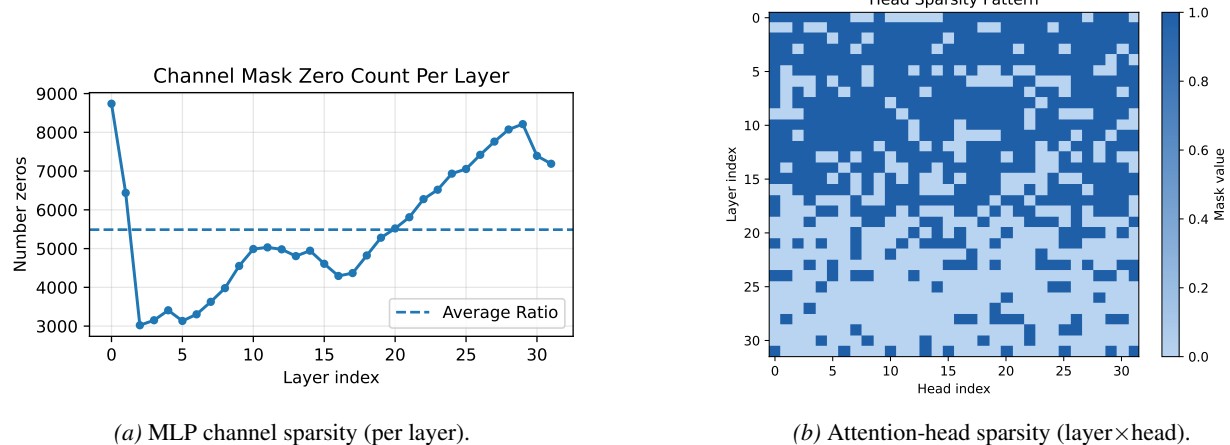

*(a)* MLP channel sparsity (per layer).

*(b)* Attention-head sparsity (layer×head).

*Figure 5.* **Dense-model sparsity patterns (LLaMA-7B, 50% sparsity).** Sparsity increases toward later layers, and head pruning becomes more selective under higher compression.

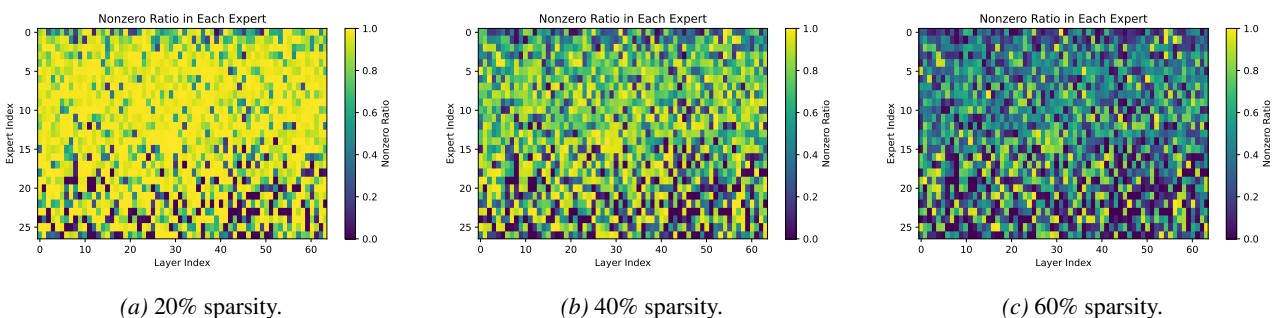

*(a)* 20% sparsity.

*(b)* 40% sparsity.

*(c)* 60% sparsity.

*Figure 6.* **MoE expert sparsity patterns (DeepSeekMoE-16B).** Expert-wise sparsity maps at different target sparsities, showing that pruning increasingly concentrates on rarely activated experts.

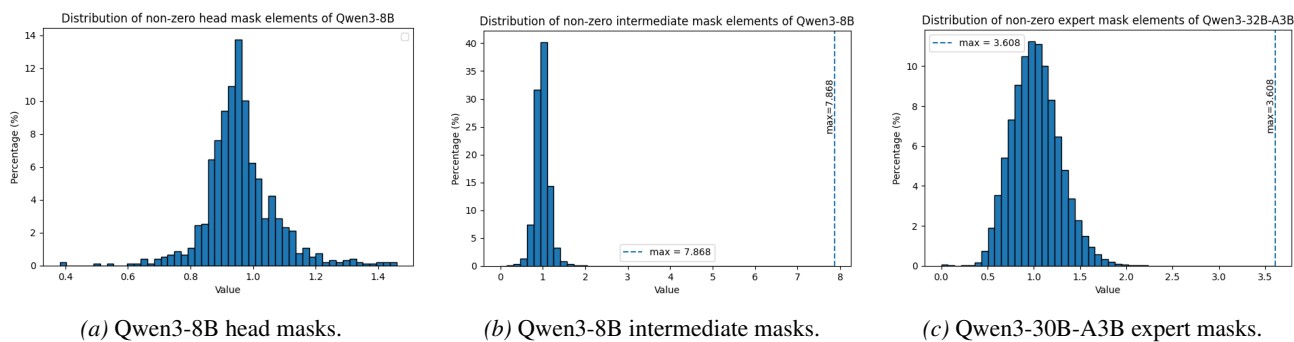

*(a)* Qwen3-8B head masks.

*(b)* Qwen3-8B intermediate masks.

*(c)* Qwen3-30B-A3B expert masks.

*Figure 7.* Magnitude distributions of learned mask values. The learned masks are centered near 1 but span a wider continuous range, demonstrating the effect of DDP's expanded mask parameterization.

