# OpenReview forum: "Deterministic Differentiable Structured Pruning for Large Language Models"
_ICML.cc/2026/Conference — ICML 2026 regular_

### Official Review · Reviewer_XJvu · 2026-02-17

**Soundness:** 4
**Presentation:** 3
**Significance:** 3
**Originality:** 3
**Overall Recommendation:** 5
**Confidence:** 4

**Summary:**

The paper proposes Deterministic Differentiable Pruning (DDP), a lightweight mask-only method for structured pruning of LLMs. It targets attention heads and MLP channels in dense models (and expert channels in MoE), using a deterministic forward gate (ReLU on logits) and an annealed smooth surrogate for l0 regularization, combined with augmented Lagrangian enforcement of a target sparsity budget and an explicit binarization loss to encourage sharp decisions. Unlike prior stochastic approaches (e.g., hard-concrete), DDP avoids sampling noise, train-test mismatch, and bounded mask ranges, leading to better expressivity and faster convergence. Experiments on LLaMA and Qwen3 models (dense and MoE) show strong results: near 1 percent loss on downstream tasks at aggressive sparsity, outperforming one-shot baselines and achieving practical speedups with vLLM.

**Compliance With Llm Reviewing Policy:**

Affirmed.

**Final Justification:**

I think this is a solid paper, backed up by a good theoretical analysis and strong empirical results.

**Key Questions For Authors:**

Could the authors add ablations (even in appendix) on the binarization loss and annealing to quantify their individual impact on convergence speed, sparsity stability, or final performance?

Do final learned forward masks (m = ReLU(z)) commonly take values much greater than 1, or do they mostly saturate near binary-like decisions after annealing? Any statistics or visualizations of mask distributions would help clarify the practical benefit of unbounded expressivity.

How sensitive is the method to the choice of $\lambda$ coefficients or annealing schedule? Were there runs where sparsity deviation exceeded 1 percent, and how were they handled?

For deployment, how exactly are nonzero masks folded into weights (e.g., scaling rows/columns)? Any minor numerical issues from tiny positive masks?

**Limitations:**

yes

**Strengths And Weaknesses:**

Strengths:
* The deterministic surrogate and decoupling of forward masks (unbounded [0, +inf)) from regularization scores ([0,1]) is a clear advance over stochastic relaxations, addressing real limitations (noise, mismatch, limited expressivity) with good motivation.
* Theoretical analysis (Theorem 3.1) provides nice guarantees on exact budget recovery under annealing and binarization.
* Empirical results are solid: consistent wins vs. strong baselines across dense/MoE families, with realistic token budgets (~30M) and measured inference gains.
* Lightweight mask-only focus makes it practical and scalable; extensions (distillation, group sparsity) integrate cleanly.

Weaknesses:
* No ablation studies isolate contributions of key components (e.g., binarization loss, annealing schedule, decoupling), so claims about faster/more stable convergence remain mostly qualitative.
* Final mask distributions are not shown or discussed (e.g., histograms of values, prevalence of intermediates or >>1), leaving uncertainty about how sharply masks binarize in practice despite the design intent.
* Slight sparsity deviation (up to 1 percent vs. target) is acknowledged but not deeply analyzed (e.g., variance across runs or sensitivity to hyperparameters).
* Comparison to stochastic mask-only methods (e.g., MaskLLM) is limited to related work; direct ablation against hard-concrete would strengthen claims.

---

> ### Author Rebuttal · Authors · 2026-03-29
>
> We thank the reviewer for the helpful suggestions. Regarding the questions:
>
> ## Weakness 1 + Question 1: Additional ablation studies and evidence for faster convergence
>
> We clarify the role of each component as follows. First, the decoupling effect is already reflected by the Det. HC column in Table 4: Det. HC removes randomness but uses the same near-binary mask for both the language modeling loss and the regularization score, while DDP decouples them, allowing a wider mask-value range. Second, we did not include a separate ablation for the binarization loss in Table 4 because, in the deterministic setting, it is crucial for driving sparse solutions; without it, mask values tend to remain near the average retention rate, making it difficult to obtain a sparse solution. For annealing, we agree that its effect should be shown more explicitly, and we provide the additional results below. For all the experiments below, we use 20% sparsity on Qwen3-8B.
>
> | Schedule              | Head Retention Ratio | Intermediate Retention Ratio | PPL  | Acc(%) |
> | --------------------- | ----------------- | ------------------------- | ---- | ------ |
> | 0.5 -> 0.05 (Default) | 0.8056            | 0.8036                    | 9.83 | 66.41  |
> | 0.5 -> 0.02           | 0.8060            | 0.8050                    | 9.87 | 66.18  |
> | 1.0 -> 0.05           | 0.8048            | 0.8029                    | 9.78 | 66.43  |
>
> These results show that DDP is not highly sensitive to the annealing schedule: all three settings achieve very similar sparsity ratios and downstream performance.
>
> To make the convergence claim more explicit, we also add a matched comparison to hard-concrete under different token budgets, showing that DDP reaches better performance with fewer tokens.
>
> | Tokens | DDP PPL | DDP Acc(%) | HC PPL | HC Acc(%) |
> | ------ | ------- | ---------- | ------ | --------- |
> | 12M    | 10.14   | 65.57      | 11.66  | 60.50     |
> | 18M    | 9.96    | 66.05      | 11.54  | 60.52     |
> | 30M    | 9.83    | 66.41      | 11.36  | 60.89     |
> | 60M    | 9.80    | 67.07      | 11.22  | 61.07     |
>
>
> ## Weakness 2 + Question 2: Final mask distributions.
>
> We agree that this is a valuable addition. We provide the distributions of the final learned mask values for the Qwen3-8B head and intermediate masks, as well as the Qwen3-32B expert masks. In all cases, the distributions are centered near 1 and are roughly symmetric. Values significantly larger than 1 appear only rarely, mainly in the intermediate and expert masks.
>
> https://anonymous.4open.science/r/Deterministic-Differentiable-Structured-Pruning-Rebuttal-3DDB/Qwen3-8B-Head.png
>
> https://anonymous.4open.science/r/Deterministic-Differentiable-Structured-Pruning-Rebuttal-3DDB/Qwen3-8B-Intermediate.png
>
> https://anonymous.4open.science/r/Deterministic-Differentiable-Structured-Pruning-Rebuttal-3DDB/Qwen3-32B-A3B-Expert.png
>
> ## Weakness 3 + Question 3: Sparsity deviation and sensitivity of $\lambda$
>
> We clarify that the small deviation arises because we do not impose a hard projection at every step. In practice, the deviation is mainly controlled by the learning rate of $\lambda_2$: a larger value reduces the deviation. We set it to keep the sparsity error within 1%; stricter sparsity matching can be achieved by increasing it. We present the sparsity ratios under different learning rates in Qwen3-8B with 20% sparsity:
>
> | LR of $\lambda_2$ | Head Retention Ratio | Intermediate Retention Ratio | PPL  | Acc(%) |
> | ----------------- | ---------- | ------------------ | ---- | ------ |
> | 16e-1             | 0.8027     | 0.8012             | 9.95 | 66.07  |
> | 8e-1 (Default)    | 0.8056     | 0.8036             | 9.83 | 66.41  |
> | 4e-1              | 0.8171     | 0.8136             | 9.78 | 66.25  |
> | 2e-2              | 0.8309     | 0.8371             | 9.90 | 66.05  |
>
> If stricter sparsity matching is needed, the learning rate of $\lambda_2$ can be increased.
>
> As for run-to-run variance, we report results across three runs with the same default hyperparameters, and the variance is limited.
>
> |       | Head Retention Ratio | Intermediate Retention Ratio | PPL  | Acc(%) |
> | ----- | ---------- | ------------------ | ---- | ------ |
> | Run 1 | 0.8064     | 0.8046             | 9.87 | 66.23  |
> | Run 2 | 0.8056     | 0.8036             | 9.83 | 66.41  |
> | Run 3 | 0.8038     | 0.8031             | 9.93 | 66.18  |
>
> ## Weakness 4: Comparison to MaskLLM.
>
> MaskLLM also uses a hard-concrete-style formulation, but is designed for 2:4 sparsity. Thus, the core methodological comparison is already captured by our HC-related ablations.
>
> ## Question 4: Final model construction and numerical issues.
>
> For MLP channels, each nonzero mask is multiplied into the corresponding row of the down-projection matrix. For attention heads, nonzero masks are multiplied into the row block corresponding to that head in O-projection matrix. Since most learned mask values are close to 1, we did not observe numerical issues in practice.

---

> > ### Author Rebuttal · Reviewer_XJvu · 2026-04-02
> >
> > My questions have been satisfactorily answered, thank you.

---

> > > ### Author Response · Authors · 2026-04-05
> > >
> > > Thank you for the positive follow-up. We are glad that our responses satisfactorily answered your questions, and we sincerely appreciate your thoughtful reading of the paper. Your comments also helped us improve the paper and clarify its presentation.

---

### Official Review · Reviewer_VEUN · 2026-03-12

**Soundness:** 4
**Presentation:** 4
**Significance:** 4
**Originality:** 3
**Overall Recommendation:** 5
**Confidence:** 4

**Summary:**

This paper studies structured pruning for large language models. Instead of fully retraining the model, the method only learns pruning masks, placing it between post-training pruning and training-based approaches.

Specifically, the authors propose a Deterministic Differentiable Pruning (DDP) method to optimize the pruning masks in a differentiable manner. Experiments on Qwen-30B show that the method achieves 20% sparsity with less than 1% accuracy drop on downstream tasks.

In addition, the pruned model is deployed with vLLM, demonstrating practical inference speedup in real serving scenarios.

**Compliance With Llm Reviewing Policy:**

Affirmed.

**Key Questions For Authors:**

1. Although the ablation study is already fairly comprehensive, it would be helpful to include experiments with knowledge distillation (KD). In particular, it is unclear whether the competing methods also leverage KD during training.
2. Could the authors visualize the layer-wise sparsity ratios and compare them with SlimLLM? Would similar sparsity patterns emerge across layers?

**Limitations:**

yes

**Strengths And Weaknesses:**

Strengths:
1. The paper is well organized, and the presentation allows readers to clearly follow the authors’ motivation, methodology, and overall workflow.
2. The proposed method reasonably leverages reparameterization and regularization techniques to improve the optimization process.
3. The experimental results demonstrate clear improvements, and the ablation study is relatively comprehensive, covering the validation of different components in the design, sparsity configurations, and the impact of training data.
4. When deployed on vLLM, the compressed model achieves tangible end-to-end inference speedups.

Weaknesses:
1. The proposed PPD method lies between training-free approaches and training-based methods, which is an important characteristic. It would be helpful to explicitly indicate in Table 2 and Table 3 whether each method requires training, to make the comparison clearer.
2. In Eq. (8), the variable $z$ appears for the first time, but its meaning is not clearly defined, which makes the equation difficult to understand.
3. In Line 222,  KKT is introduced without providing its full name when it first appears.

---

> ### Author Rebuttal · Authors · 2026-03-29
>
> We thank the reviewer for the positive assessment and constructive suggestions. Regarding your questions:
>
> ## Weakness 1: Comparisons in Tables 2–3 would be clearer if each method were labeled by training type.
>
> We agree. In the revision, we will explicitly annotate the optimization regime of each method in Tables 2–3. For dense baselines, LoRAPrune, LoRAP, and SlimLLM will be labeled as one-shot pruning followed by post-pruning LoRA fine-tuning, while DDP will be labeled as mask-only optimization with frozen backbone weights. For MoE baselines, NAEE, D$^2$-MoE, Camera-P, and HEAPr will be labeled as training-free methods, while DDP will again be labeled as mask-only optimization. This will make the comparison clearer and better position DDP as a lightweight middle ground between training-free pruning and post-pruning weight-updating methods.
>
> ## Weakness 2-3: Some presentation details need clarification.
>
> We agree and will fix these in the revision by defining $z$ explicitly at its first appearance in Eq. (8) and by spelling out Karush–Kuhn–Tucker (KKT) at first mention.
>
> ## Question 1: Clarification of the role of knowledge distillation (KD).
>
> Thank you for raising this point. KD is not required for DDP, but it is a useful optional component that can further improve performance. As shown in our component ablation in Table 4, adding KD on top of deterministic mask optimization and the expanded forward-mask parameterization yields the best results in both dense and MoE settings. That said, the gains from KD are secondary; the primary advantage of DDP comes from the deterministic mask-learning formulation itself rather than from distillation alone.
>
> For the competing baselines in Tables 2–3, we did not include KD in order to follow their original setup. We also note that some baselines are less naturally compatible with KD, since they apply nontrivial transformations to the dense model and therefore require an additional dense copy to serve as the teacher. In contrast, our mask-only optimization shares the same frozen backbone weights across both passes, so incorporating KD does not require an extra model, additional trainable backbone parameters, or optimizer states.
>
> To further address this point, we additionally evaluate KD on a compatible baseline, LoRAP, under the same experimental setting on LLaMA-2-7B with 20% sparsity. As shown below, KD consistently improves LoRAP over its original setup. Nevertheless, DDP still achieves stronger performance without any weight updates, suggesting that the main gain comes from the proposed mask-learning formulation rather than from KD alone.
>
> |                   | Wiki PPL | Mean Acc |
> | ----------------- | -------- | -------- |
> | LoRAP             | 15.02    | 59.44    |
> | LoRAP+finetune    | 14.67    | 61.24    |
> | LoRAP+finetune+KD | 14.58    | 61.86    |
> | Ours              | 14.39    | 64.82    |
>
> ## Question 2: Layer-wise sparsity visualization and sparsity patterns across layers.
>
> We agree that layer-wise sparsity visualization is important, as it helps reveal whether DDP learns similar broad patterns to heuristic methods or departs from them in ways that explain its performance gains. We already include partial results on the sparsity distribution of MLP channels and attention heads in the original paper. Because SlimLLM does not release the exact allocation implementation, we re-implemented its layer-allocation heuristic based on the cosine-similarity description in the paper and provide a direct comparison of layer-wise MLP sparsity for LLaMA-2-7B and Qwen3-8B below. Anonymous figures are provided here:
>
> https://anonymous.4open.science/r/Deterministic-Differentiable-Structured-Pruning-Rebuttal-3DDB/LLaMA2-7B.png
>
> https://anonymous.4open.science/r/Deterministic-Differentiable-Structured-Pruning-Rebuttal-3DDB/Qwen3-8B.png
>
> The results show a similar coarse trend, with model-dependent differences and generally higher sparsity in deeper layers, but DDP exhibits greater cross-layer variation and thus a more selective allocation pattern. We believe this reflects the advantage of optimization-based mask learning: unlike heuristic allocation, it can better capture the complex interactions across layers, heads, and channels, which likely contributes to the stronger empirical performance.
>
> Overall, we appreciate the reviewer’s strong support. The requested changes are mainly about clarity and interpretability, and we believe they can be addressed cleanly in the revision.

---

> > ### Author Rebuttal · Reviewer_VEUN · 2026-04-02
> >
> > My concerns have been addressed, and I will keep my positive score unchanged.

---

> > > ### Author Response · Authors · 2026-04-05
> > >
> > > Thank you for the positive assessment and for taking the time to review our rebuttal. We are glad that our responses addressed your concerns, and we sincerely appreciate your continued support. Your comments were also helpful in improving the paper and clarifying its presentation.

---

### Official Review · Reviewer_mniz · 2026-03-13

**Soundness:** 3
**Presentation:** 3
**Significance:** 2
**Originality:** 3
**Overall Recommendation:** 4
**Confidence:** 4

**Summary:**

This paper presents a new algorithm for structured LLM pruning. It is based on an augmented Lagrangian method combined with a deterministic smoothing approach for the $L_0$ norm. Starting from the common probabilistic smoothing approach (hard-concrete relaxation), the paper proposes a deterministic alternative and presents theoretical results. The method is then compared numerically with related work on LLM pruning for models up to 30B parameters and provides extensive ablations of different variants and hyperparameters.

**Compliance With Llm Reviewing Policy:**

Affirmed.

**Final Justification:**

I am still not convinced that the complexity of this approach is necessary as opposed to simpler classical approaches from sparse optimization, but I have to conceide that this method is effective as demonstrated by the experiments and so I raised my score to 4 (weak accept).

**Key Questions For Authors:**

See Strengths And Weaknesses.

**Limitations:**

yes

**Strengths And Weaknesses:**

The paper proposes an interesting new method with some theoretical justification and demonstrates with extensive empirical experimens that it outperforms several prior works on multiple architectures and sparsity levels.

The main weakness of this paper is that it introduces a relatively complicated method for sparse optimization, which is a problem that has been studied extensively both in the classical optimization literature and in more recent LLM pruning literature.

In particular, given the mask-based formulation used in this paper (where only multiplicative mask variables are optimized while the model weights remain frozen), several classical sparse optimization methods could potentially be applied directly to the same formulation. Examples include:

* ISTA (Iterative Shrinking-Thresholding Algorithm), here the relaxation would to go from $L_0$ to $L_1$
* ADMM (Alternating Direction Method of Multipliers)
* PGD (Projected Gradient Descent) directly on the $L_0$ ball (the set is non-convex, but the projection is still available in closed form)

Comparing to such baselines, or discussing why they are not applicable in this setting, seems necessary to motivate the use of a relatively complex algorithm such as DDP.

**Regarding the experiments:**

* You state that a single configuration is used across all runs. It is unclear to me whether the baselines are tuned equivalently. Details such as LoRA rank, learning rate, and training schedules for the baselines are not reported.
* In Section 5.3.3, the impact of the token budget is ablated for DDP. However, several baselines involve post-pruning fine-tuning, which may benefit significantly from larger token budgets.
* For MoE models, the comparison is limited to training-free pruning methods, while DDP uses data. It would be helpful to compare against methods that perform post-pruning recovery (e.g., LoRA or continued training) to ensure that the advantage does not stem primarily fromusing data.
* You state that sparsity may deviate by up to 1% from the target. Since the performance differences between methods are sometimes small, such deviations could influence the ranking of methods. It would improve clarity to report the exact achieved sparsity for each experiment or enforce identical sparsity levels across methods.


I would be willing to raise my score to accept if the authors can demonstrate that the method clearly outperforms simpler baseline methods I mentioned.

---

> ### Author Rebuttal · Authors · 2026-03-31
>
> We thank the reviewer for your suggestion, regarding your questions:
>
> ## Weakness 1: Comparison with classical sparse optimization methods.
>
> Our setting is not generic sparse optimization, but structured mask-only optimization with explicit target sparsity. The main goal is to **drive unwanted masks to exact zero while avoiding unnecessary shrinkage of the retained continuous mask values**. We clarify below why the methods mentioned by the reviewer do not align well with this objective.
>
> - For ISTA, the natural counterpart is an l1-relaxed formulation. Its main limitation is that plain l1 decay applies uniform shrinkage to all elements. In our setting, this severely over-shrinks the surviving continuous masks and leads to very poor performance (>100 PPL on LLaMA-2-7B). It also does not directly enforce an exact keep budget, so matching a target sparsity requires additional calibration of the regularization strength.
>
> - For PGD, we implemented a hard top-k projected baseline. Since all mask values are initialized to 1, the top-k support is largely determined by the first few gradient updates. This causes early support freezing: once a support is selected, the mask becomes effectively fixed. We found this to be a practical issue in our setting, and it again produced suboptimal results.
>
> - For ADMM, we adapted it to our setting by introducing a continuous mask variable and a projected sparse auxiliary variable, coupled through a consensus penalty. However, for the same reason as PGD, the first few projection steps tend to select the support based only on very small early gradient differences, and the consensus penalty then keeps pulling the continuous mask toward that same support. Moreover, ADMM is more complex to implement in our setting as it requires additional variables and a separate update schedule.
>
> In contrast, DDP decouples forward mask values from sparsity control, which avoids unnecessary decay of retained mask values and allows a wider range of continuous mask values. The augmented-Lagrangian objective together with the binarization term separates useful and redundant masks directly through the loss, rather than through heuristic magnitude-based selection, which leads to substantially better results.
>
> ## Weakness 2: Baseline tuning details.
>
> We clarify our design as follows: for baselines with public implementations, we start from the settings reported in the original papers and performe limited tuning over the main optimization hyperparameters, but did not observe material gains. For baselines without public code, we reported the published numbers directly and will label them explicitly.
>
> ## Weakness 3: Recovery-based baselines on more tokens.
>
> Our current comparison intentionally keeps DDP at similar cost to the baselines, rather than stacking additional recovery. As shown in our response to Reviewer d6kJ (Limitation 2), DDP is fully compatible with LoRA or full fine-tuning; thus, the current results should be interpreted as the quality of the pruned initialization produced by DDP. We additionally tested LoRA recovery with increasing token budgets on LLaMA-2-7B at 20% sparsity using LoRAP as the baseline. Although recovery improves performance steadily, it remains below DDP without extra recovery even at 120M tokens. **This suggests that DDP already provides a substantially stronger starting point, while remaining compatible with further recovery.**
>
> | Tokens    | Wiki PPL | Mean Acc |
> | --------- | -------- | -------- |
> | 0M        | 15.02    | 59.44    |
> | 15M       | 14.94    | 60.86    |
> | 30M       | 14.67    | 61.24    |
> | 60M       | 14.49    | 62.77    |
> | 120M      | 14.46    | 62.98    |
> | DDP (Ours) | 14.39    | 64.82    |
>
> ## Weakness 4: MoE models and data-based recovery.
>
> We agree this should be clarified. To address this directly, we additionally evaluated HEAPr on DeepSeek-MoE at 20% sparsity with LoRA recovery (rank 8), using the same 30M-token data budget as our method. Even with this additional recovery on the same data, it remains below our mask-only method:
>
> |                  | C4 PPL | Mean ACC |
> | ---------------- | ------ | -------- |
> | Dense            | 9.05   | 62.28    |
> | HEAPr            | 9.85   | 60.62    |
> | HEAPr + LoRA | 9.68   | 60.79    |
> | DDP(Ours)        | 9.38   | 61.84    |
>
> ## Weakness 5: Exact achieved sparsity.
>
> We show representative LLaMA-2-7B results below; we will add the full results in the revision. In practice, the achieved sparsity remains very close to the target.
>
> | Target Sparsity | Achieved Head Sparsity | Achieved Intermediate Sparsity |
> | --------------- | ---------------------- | ------------------------------ |
> | 20%             | 20.31%                 | 19.73%                         |
> | 50%             | 49.37%                 | 49.72%                         |
>
> We hope this response addresses your concerns, and we would be happy to answer any further follow-up questions

---

> > ### Author Rebuttal · Reviewer_mniz · 2026-04-04
> >
> > Thank you for your thorough review.
> >
> > Your answers Weakness 2-5 fully resolve my concerns, but I am not convinced by your answer regarding Weakness 1.
> >
> >
> > Regarding ISTA:
> > I agree that the regularization parameter has to be tuned, but your method also introduces additional hyperparameters, such as $\lambda_1, \lambda_2, \lambda_3, \mu_t$. Also your approach does not allow to control the precise sparsity either, so this issues is not specific to ISTA.
> >
> > Regarding ADMM/PGD:
> > I understand this issue that the first few gradients determine the sparsity, but similar problems would probably happen in your method if you didn't anneal $\mu_t$. I assume that this problem could be at least partially fixed by slowly ramping up the sparsity.
> >
> > I appreciate your effort to implement these baselines, but it seems to me that the issues you note could be quite easily circumvented by tricks very similar to those you used for your method. I do not claim that these methods would perform better than your approach, but your rebuttal does not convince me that your approach, which seems much more complex than these baselines has a fundamental advantage. Therefore, I will keep my score.

---

> > > ### Author Response · Authors · 2026-04-05
> > >
> > > We thank the reviewer for confirming that Weaknesses 2–5 are resolved. Regarding Weakness 1, in our earlier response we did not state this point sharply enough due to space constraints. We note that, to our knowledge, there is no prior evidence in the literature that these classical methods, when adapted to our exact setting—mask-only optimization with all masks initialized to 1 and an targeted final sparsity ratio—can achieve the same design goals. More importantly, **our point is not merely that such adapted baselines are empirically suboptimal, but that they have fundamental drawbacks in this setting that limit their performance.** Below, we identify four properties that are crucial in our setting:
> > >
> > > **Property 1.**  Mask selection should be driven directly by the training loss, rather than handcrafted heuristics such as magnitude, since such heuristics can be inaccurate and may not align with the true downstream objective. This is well established in the weight pruning literature: simple magnitude rules are inferior to loss-aware criteria such as SparseGPT.
> > >
> > > **Property 2.** Optimization should follow a continuous, differentiable path without repeated discrete changes in the sparsity pattern, since manually imposing abrupt, discontinuous mask updates during training can destabilize optimization.
> > >
> > > **Property 3.** The mask parameterization should remain expressive: after a subset of components is driven to zero, the remaining mask values should vary freely in a flexible range, rather than being compressed into a near-binary interval or uniformly shrunk by the regularizer. This requires a sufficiently discriminative sparsity mechanism that can eliminate redundant components without unnecessarily suppressing retained ones.
> > >
> > > **Property 4.** Training should be fully deterministic, to avoid the train-test mismatch and optimization noise introduced by random mask sampling.
> > >
> > > In the paper, we already explained why hard-concrete-style methods do not satisfy Properties 3 and 4. Here we further clarify why the classical alternatives mentioned by the reviewer do not satisfy all four properties simultaneously.
> > >
> > > - **ISTA:** The natural L1-relaxed alternative enforces sparsity through uniform shrinkage of the mask values themselves, so the same mechanism that zeros redundant components also suppresses retained ones, conflicting with Property 3. In our experiments, the average retained mask value dropped to 0.15 at 20% sparsity on LLaMA-2-7B, whereas DDP preserves a much richer spread of retained mask values (see our response to Reviewer XJvu). Adding thresholding or selective decay could reduce this effect, but then support selection depends on an explicit magnitude heuristic, conflicting with Property 1.
> > >
> > > - **PGD:** PGD enforces exact sparsity through hard projection, so optimization necessarily undergoes repeated non-smooth support changes, which conflicts with Property 2. Moreover, the projection is based solely on mask magnitude and therefore introduces a heuristic support-selection rule, conflicting with Property 1. While gradually increasing sparsity, as the reviewer suggests, may partially alleviate the early support-freezing issue, it does not remove the fundamental reliance on repeated hard, magnitude-based support updates. Our new experimental results show that a linear sparsity ramp performed worse than the naive PGD baseline, consistent with the instability caused by frequent support changes.
> > >
> > > - **ADMM:** In ADMM, exact sparsity is still enforced by hard projection of an auxiliary variable, so it likewise relies on repeated non-smooth, magnitude-based support updates, conflicting with both Properties 1 and 2. Compared with PGD, it also introduces additional optimization machinery, including auxiliary and dual variables and an alternating update schedule, making it more complex without resolving the core issue.
> > >
> > > **Summary:** Our method instead starts from an L0-constrained, loss-driven objective with ALM, which enforces the target sparsity through the training objective itself rather than heuristic support selection. The surrogate mapping and binarization loss make the sparsity mechanism sufficiently discriminative to drive redundant masks to zero without unnecessarily shrinking retained mask values. The resulting procedure is still simple to implement: a deterministic surrogate optimization with no stochastic sampling, preserving a smooth differentiable optimization path. For this reason, we view our modification as **nontrivial and fundamental**: it is not a minor implementation variant of existing sparse optimization routines, but a different formulation designed to satisfy these properties simultaneously. We believe this is the key reason it consistently achieves stronger empirical results, and why the same combination of properties is not recovered by minor practical adjustments to prior baselines.
> > >
> > > We hope this clarifies our point, and we would be happy to provide any further clarification if helpful.

---

### Official Review · Reviewer_YFVQ · 2026-03-17

**Soundness:** 3
**Presentation:** 3
**Significance:** 2
**Originality:** 2
**Overall Recommendation:** 4
**Confidence:** 4

**Summary:**

The submission proposes a strategy for optimizing structured pruning in LLMs.  It is based on a Lagrangian relaxation, which is intended to more closely match training and deployment, and some informal analysis supports the strategy.  Asymptotic convergence results in the appendex show that the strategy will eventually recover the correct solution.  Experimental results in Tables 2 & 3 show marginal improvement on LLaMA and DeepSeek models.

**Compliance With Llm Reviewing Policy:**

Affirmed.

**Final Justification:**

The author rebuttal, particularly on "weakness 2" and "limitation 1," was appreciated.  I have upgraded my score from a 3 to a 4.

**Key Questions For Authors:**

Do you have theoretical claims for the improvement over competing methods based on the hard-concrete relaxation?

**Limitations:**

The method is primarily empirically validated on a limited set of LLMs coming from two main famlies.  Generalization to other families is likely but unproven.

**Strengths And Weaknesses:**

Strategy is sensible and the Lagrangian formulation is a natural one for the constrained optimization problem.

No convergence rates for optimization theory are given to analyze the claimed improvements.  Empirical results are marginal over existing methods.

---

> ### Author Rebuttal · Authors · 2026-03-29
>
> We thank the reviewer for the careful assessment. We are encouraged that the reviewer finds the Lagrangian formulation natural for constrained pruning. Below we clarify the theoretical scope, the empirical results, and the model-family coverage.
>
> ## Weakness 1 + Question 1:  Lack of convergence rates and theoretical improvement.
>
> We would like to clarify that this paper is primarily about proposing a practical, lightweight structured pruning method rather than establishing a complete non-asymptotic convergence theory for model pruning. DDP is a mask-only approach that is inexpensive to optimize, improves the quality-efficiency tradeoff across both dense and MoE LLMs, and delivers real deployment value. In particular, DDP incurs only modest mask-learning cost relative to training-based approaches while achieving practical vLLM speedups with minimal performance loss in deployment.
>
> We do not claim a formal non-asymptotic rate improvement over hard-concrete. Instead, our theoretical contribution is to show that DDP admits a deterministic constrained surrogate with convergence to feasible/KKT points and exact-budget recovery. Specifically, DDP defines a fully deterministic constrained surrogate whose accumulation points are feasible and first-order stationary, and annealing plus binarization recover the exact hard keep budget. By contrast, prior hard-concrete methods optimize sampled masks during training but deploy deterministic masks, introducing sampling noise and train-test mismatch, without offering a comparable convergence guarantee in the mask-only setting. For a paper focused on practical structured pruning with clear empirical and deployment gains, we believe establishing convergence and exact-budget recovery is an appropriate level of theoretical support.
>
> ## Weakness 2: Empirical improvement over existing methods.
>
> We respectfully disagree that the gains are marginal. Since mean accuracy is averaged over a mixed benchmark suite, low-sparsity differences can appear compressed in the aggregate. Moreover, DDP shows clearer gains on harder tasks: on LLaMA-7B, it outperforms SlimLLM on HellaSwag by +1.97 at 20% sparsity and +5.63 at 50% sparsity. More importantly, the advantage widens as pruning becomes more aggressive. For example, on LLaMA-13B at 50% sparsity, DDP reaches 62.14 mean accuracy versus 58.20 for LoRAP and 54.75 for SlimLLM; on DeepSeekMoE-16B at 60% sparsity, it improves mean accuracy from 51.62 to 58.18 while reducing perplexity from 18.10 to 12.65. We will revise the paper to emphasize that DDP’s main empirical advantage is its stronger robustness on harder tasks and under aggressive pruning.
>
> ## Limitation 1: Extension to more model families.
>
> We clarify that the original submission already evaluates DDP on the main dense and MoE model families used in the most relevant prior baselines, rather than on a narrow set of architectures. To further reinforce this point, we now add results on the Mistral family, including Mistral-7B-v0.3 and Mistral-Nemo. The same pattern holds as in the main paper: DDP remains competitive at low sparsity and shows a substantially larger advantage at higher sparsity. At 50% sparsity, DDP improves mean accuracy from 46.21 to 50.87 on Mistral-7B-v0.3 and from 47.92 to 53.53 on Mistral-Nemo, while also reducing Wiki perplexity from 15.53 to 9.88 and from 16.85 to 12.07, respectively. This provides further evidence that DDP generalizes beyond the model families originally reported.
>
> | Model           | Metric   | Method         | 12.5% |   25% | 37.5% |   50% |
> | --------------- | -------- | -------------- | ----: | ----: | ----: | ----: |
> | Mistral-7B-v0.3 | Wiki PPL   | Týr-the-Pruner |  5.61 |  7.08 | 10.25 | 15.53 |
> |                 |          | DDP            |  5.59 |  6.73 |  7.80 |  9.88 |
> |                 | Mean Acc | Týr-the-Pruner | 63.05 | 60.22 | 52.34 | 46.21 |
> |                 |          | DDP            | 66.31 | 62.85 | 58.39 | 50.87 |
> | Mistral-Nemo    | Wiki PPL    | Týr-the-Pruner |  6.31 |  7.87 | 11.47 | 16.85 |
> |                 |          | DDP            |  6.25 |  7.55 |  9.39 | 12.07 |
> |                 | Mean Acc | Týr-the-Pruner | 64.15 | 60.61 | 54.63 | 47.92 |
> |                 |          | DDP            | 67.23 | 63.99 | 58.95 | 53.53 |
>
> If our response has addressed your concerns, we hope you will consider updating your score accordingly. We would also be happy to answer any further follow-up questions.

---

> > ### Author Rebuttal · Reviewer_YFVQ · 2026-04-02
> >
> > The clarification in claim of where the improvements are to be expected helps to interpret the potential value of the contribution.  Additional results on Mistral models is also appreciated.

---

> > > ### Author Response · Authors · 2026-04-05
> > >
> > > Thank you for the thoughtful follow-up and for the positive assessment. We are glad that the clarification on where the improvements should be expected helped make the contribution easier to interpret, and we also appreciate your recognition of the additional Mistral results. Your comments were very helpful in improving the paper and sharpening how we present its value.

---

### Official Review · Reviewer_d6kJ · 2026-03-24

**Soundness:** 3
**Presentation:** 3
**Significance:** 3
**Originality:** 3
**Overall Recommendation:** 4
**Confidence:** 3

**Summary:**

This paper presents Deterministic Differentiable Pruning (DDP), a lightweight mask-only framework for accelerating structured pruning of dense and MoE LLMs.
DDP formulates pruning as an ℓ₀-constrained optimization problem, replacing stochastic hard-concrete relaxations with an annealed deterministic soft surrogate while decoupling the ReLU forward mask ([0, ∞)) from the retention-score regularizer. Guided by augmented Lagrangian penalties and an explicit binarization loss, it enforces exact per-layer/global budgets, eliminating train–test mismatch and sampling noise.
Experiments on LLaMA, Qwen3, and DeepSeekMoE models (up to 32B) show DDP outperforms prior baselines at 20–60% sparsity with ≤1% quality loss and delivers 1.36–2.2× vLLM speedups.

**Compliance With Llm Reviewing Policy:**

Affirmed.

**Final Justification:**

This paper demonstrates clear novelty and is supported by solid experimental validation of the proposed method’s effectiveness.  And all concerns have been addressed, thus I keep my positive rating.

**Key Questions For Authors:**

1.The authors should provide pruning results on significantly larger LLMs (e.g., 70B+ models ) to better demonstrate the generalization capability and scalability of DDP.

2.The paper lacks a systematic comparison of the additional trainable parameters introduced by the proposed pruning method (across different model sizes) with those of other similar mask-based or structured pruning baselines, along with quantitative analysis of computational costs such as peak memory usage, optimizer memory overhead, and total training FLOPs.

3. Additional comparisons against more recent state-of-the-art (SOTA) structured pruning methods (e.g., Týr-the-Pruner) would further strengthen the positioning of DDP’s contributions.

**Limitations:**

1.To strengthen the evaluation of DDP’s efficiency, the authors should include a systematic comparison of the additional trainable parameters introduced by the proposed method (across different model sizes) against those of other mask-based or structured pruning baselines, together with quantitative analysis of computational costs such as peak memory usage, optimizer memory overhead, and total training FLOPs.

2.The authors should discuss or empirically explore whether DDP (as a pure mask-only approach) can be combined with weight-updating pruning schemes (e.g., LoRA-based fine-tuning or continued full-parameter training) to further improve pruning quality and close the accuracy gap under higher sparsity ratios.

**Strengths And Weaknesses:**

Strengths:
1.This work is a mask-only setting to completely eliminate stochasticity, directly solving train–test mismatch and slow convergence while expanding mask expressivity from [0,1] to [0, ∞) via decoupled ReLU forward masks.

2. To address the non-differentiability of the ℓ₀ norm and enforce precise sparsity budgets, the paper introduces an annealed deterministic smooth surrogate, augmented Lagrangian method (ALM), and explicit binarization loss, enabling stable gradient-based optimization with theoretical guarantees of KKT convergence and exact budget recovery.

3.The paper includes a rich set of figures (Fig. 1–3), tables, and code-level details (Algorithm 1) that clearly illustrate the method workflow, surrogate mapping, and sparsity patterns, together with comprehensive ablations and support for dense/MoE models across global, layer-wise, and expert-wise granularities, greatly improving readability and reproducibility.

4.Experiments on flagship models (LLaMA, Qwen3, DeepSeekMoE up to 32B) demonstrate strong engineering value: lightweight 30M-token training (~20 min on 7B models), vLLM end-to-end speedups of 1.36–2.2×, exceptional robustness on MoE expert pruning, and consistent SOTA quality–efficiency trade-offs (≤1% loss) without any post-pruning fine-tuning.


Weaknesses:

1. The experimental scale remains relatively limited (Qwen3-32B appears to be the largest model evaluated, with no results reported on 70B+ models). In addition, comparisons against the latest state-of-the-art methods—such as Týr-the-Pruner (“Unlocking Accurate 50% Structural Pruning for LLMs via Global Sparsity Distribution Optimization”)—are entirely missing.

2.The paper lacks a systematic comparison of the additional trainable parameters introduced by DDP across different model sizes against those required by similar mask-based or structured pruning baselines, as well as any quantitative analysis of other computational costs (e.g., peak memory usage, optimizer memory, or total training FLOPs).

---

> ### Author Rebuttal · Authors · 2026-03-30
>
> We thank the reviewer for the constructive feedback and for recognizing the technical clarity and practical value of DDP. Regarding the concerns:
>
> ## Weakness 1 + Question 3: Comparison with additional baselines
>
> We provide a direct comparison with Týr-the-Pruner below. Under matched sparsity levels, DDP consistently achieves better WikiText perplexity and mean zero-shot accuracy on both LLaMA-2-7B and LLaMA-2-13B. Additional comparisons on Mistral models are provided in our response to Reviewer YFVQ.
>
> | Model      | Metric   | Method         | 12.5% |   25% | 37.5% |   50% |
> | ---------- | -------- | -------------- | ----: | ----: | ----: | ----: |
> | LLaMA-2-7B  | Wiki     | Týr-the-Pruner |  5.84 |  7.51 | 10.29 | 16.17 |
> |            |          | DDP            |  5.72 |  6.87 |  8.17 | 10.34 |
> |            | Mean Acc | Týr-the-Pruner | 56.98 | 54.64 | 52.21 | 47.41 |
> |            |          | DDP            | 61.99 | 60.29 | 55.98 | 52.17 |
> | LLaMA-2-13B | Wiki     | Týr-the-Pruner |  5.03 |  5.79 |  7.17 |  9.59 |
> |            |          | DDP            |  5.03 |  5.75 |  6.65 |  8.14 |
> |            | Mean Acc | Týr-the-Pruner | 62.66 | 61.16 | 58.67 | 54.58 |
> |            |          | DDP            | 65.62 | 64.51 | 60.88 | 58.43 |
>
> ## Weakness 1 + Question 1: Limited experimental scale.
>
> We agree that results on 70B+ models would further strengthen the paper. At the same time, the current paper already covers models up to Qwen3-32B and Qwen3-30B-A3B, and our additional dense Qwen3 results show a clear scaling trend: larger models are consistently less lossy at the same pruning ratio. A controlled 70B+ evaluation is beyond the scope of the rebuttal period because it requires substantially more compute, matched sparsity sweeps, baseline runs, and full evaluation.
>
> ## Weakness 2 + Question 2 + Limitation 1: Comparison of the additional trainable parameters and computational cost.
>
>
> We provide a matched comparison between DDP, DDP with distillation, LoRA fine-tuning, and full-parameter fine-tuning on LLaMA-2-7B with distributed data-parallel training, per-device batch size 2, sequence length 2,048, and activation checkpointing. Under this setup, activation memory is similar across methods, so the main difference comes from trainable-state overhead. DDP uses orders of magnitude fewer trainable parameters than LoRA and full fine-tuning, while keeping per-step cost comparable to LoRA; without distillation, it is about 17% faster than LoRA because it adds only lightweight layerwise mask operations. Adding distillation increases cost modestly due to the extra teacher forward pass, but the overall cost remains far below full fine-tuning. Full fine-tuning is substantially more expensive due to much larger gradient and optimizer memory, as well as higher per-step FLOPs and wall-clock time. This further supports our claim that DDP is a lightweight and scalable mask-only alternative to weight-updating pruning.
>
> | Method        | Trainable Params | Activation Mem | Grad/Opt Mem | Model Mem | Total FLOPs | Time / step |
> | ------------- | ---------------- | -------------- | ------------ | --------- | ----------- | ----------- |
> | DDP           | 0.35M            | 5 GB           | negligible   | 13 GB     | 0.97E       | 1.5 s       |
> | DDP (distill) | 0.35M            | 6 GB           | negligible   | 13 GB     | 1.254E      | 1.9 s       |
> | LoRA (r = 8)  | 20M              | 5 GB           | small        | 13 GB     | 0.97E       | 1.8 s       |
> | Full FT       | 7B               | 5 GB           | 65 GB        | 13 GB     | 1.40E       | 2.4 s       |
>
> ## Limitation 2: Combination with weight-updating recovery.
>
> We agree that combining DDP with weight-updating recovery is an important direction. Since DDP produces a pruned model directly, it can be naturally followed by further weight updates. To examine this, we additionally apply LoRA fine-tuning (rank 8) and full-parameter fine-tuning for 30M tokens on the 20%-sparse LLaMA-2-7B model. As shown below, both settings further improve the pruned model, with especially clear gains in mean zero-shot accuracy, substantially narrowing the gap to the dense baseline. These results suggest that DDP is compatible with standard post-pruning recovery and can serve as a lightweight first stage before subsequent weight updating.
>
> |                     | Wiki PPL | Mean Acc |
> | ------------------- | -------- | -------- |
> | Dense               | 12.18    | 66.63    |
> | DDP                 | 14.39    | 64.82    |
> | DDP + LoRA FT | 14.20    | 65.57    |
> | DDP + Full FT | 14.12    | 65.89    |
>
> If our response has addressed your concerns, we hope you will consider updating your score accordingly. We would also be happy to answer any further follow-up questions.

---

> > ### Author Rebuttal · Reviewer_d6kJ · 2026-04-01
> >
> > All concerns have been addressed, thus I keep my positive rating.

---

> > > ### Author Response · Authors · 2026-04-05
> > >
> > > Thank you for the positive assessment and for taking the time to read our rebuttal carefully. We are glad the clarifications addressed your concerns, and we sincerely appreciate your encouraging evaluation and continued support. Your comments and suggestions were also very helpful in improving the paper and sharpening its presentation.

---

### Decision · Program_Chairs · 2026-04-30

**Decision:**

Accept (regular)

**Comment:**

This paper proposed Deterministic Differentiable Pruning (DDP), a method for structured pruning of large language models (LLMs) using masks and L0 sparsity constraint. Because L0 norm is not differentiable everywhere, prior work typically adopts stochastic hard-concrete relaxations to enable differentiable optimization; however, this stochasticity can introduce a train–test mismatch when sampled masks are discretized for deployment and restricts masks to a bounded, near-binary range. Instead, the proposed DDP method used reparameterization trick and introduced annealed deterministic L0 soft surrogate and not confining the mask values to a very bounded range for optimization. The paper showed that DDP can achieve better performance loss than previous methods at the same sparsity level for various types of dense and MoE models of different sizes.

Despite there are debates between one of the reviewers and the authors on whether such a complex surrogate for pruning is necessary, all the reviewers agreed that the analysis and evaluation are solid and the empirical results are quite impressive. Because all the reviewers scored positively on the paper, I would like to recommend this paper for acceptance.